# Insights into the molecular architecture and histone H3-H4 deposition mechanism of yeast Chromatin assembly factor 1

Paul Victor Sauer[1], Jennifer Timm[1], Danni Liu[2,3], David Sitbon[4,5], Elisabetta Boeri-Erba[6,7], Christophe Velours[3], Norbert Mücke[8], Jörg Langowski[8], Françoise Ochsenbein[2,3], Geneviève Almouzni[4,5], Daniel Panne[1]*

[1]European Molecular Biology Laboratory, Grenoble, France; [2]CEA, DRF, SB2SM, Laboratoire de Biologie Structurale et Radiobiologie, Gif-sur-Yvette, France; [3]Institute for Integrative Biology of the Cell (I2BC), CEA, CNRS, Université Paris-Sud, Gif-sur-Yvette, France; [4]Institut Curie, PSL Research University, CNRS, UMR3664, Equipe Labellisée Ligue contre le Cancer, Paris, France; [5]Sorbonne Universités, UPMC Univ Paris 06, CNRS, UMR3664, Paris, France; [6]Université Grenoble Alpes, Institut de Biologie Structurale (IBS), Grenoble, France; [7]Commissariat à l'Energie Atomique et aux Energies Alternatives (CEA), Grenoble, France; [8]Abteilung Biophysik der Makromoleküle, Deutsches Krebsforschungszentrum, Heidelberg, Germany

*For correspondence: panne@embl.fr

**Abstract** How the very first step in nucleosome assembly, deposition of histone H3-H4 as tetramers or dimers on DNA, is accomplished remains largely unclear. Here, we report that yeast chromatin assembly factor 1 (CAF1), a conserved histone chaperone complex that deposits H3-H4 during DNA replication, binds a single H3-H4 heterodimer in solution. We identify a new DNA-binding domain in the large Cac1 subunit of CAF1, which is required for high-affinity DNA binding by the CAF1 three-subunit complex, and which is distinct from the previously described C-terminal winged-helix domain. CAF1 binds preferentially to DNA molecules longer than 40 bp, and two CAF1-H3-H4 complexes concertedly associate with DNA molecules of this size, resulting in deposition of H3-H4 tetramers. While DNA binding is not essential for H3–H4 tetrasome deposition in vitro, it is required for efficient DNA synthesis-coupled nucleosome assembly. Mutant histones with impaired H3-H4 tetramerization interactions fail to release from CAF1, indicating that DNA deposition of H3-H4 tetramers by CAF1 requires a hierarchical cooperation between DNA binding, H3-H4 deposition and histone tetramerization.

## Introduction

Nucleosomes in eukaryotic cells enable packaging of the DNA within the cell nucleus and provide an important layer in genome regulation. They are composed of an octameric core of histones, around which 147 bp of DNA are wrapped (*Luger et al., 1997*). The majority of nucleosomes in chromatin contain two copies of each of the four major histones H2A, H2B, H3 and H4 that are assembled in a step-wise manner (*Smith and Stillman, 1991*). Following the initial H3-H4 tetramer, two H2A-H2B dimers are deposited to complete the octameric core particle. Nucleosome assembly is promoted via the action of histone chaperones, (*De Koning et al., 2007*; *Gurard-Levin et al., 2014*).

DNA replication requires doubling of the amount of chromatin which is accomplished through recycling of parental and incorporation of newly synthesized histones (*Groth, 2009*). It has been

**eLife digest** Animal and plant cells contain very long DNA molecules that are tightly packaged by being wrapped around proteins called histones to form structures known as nucleosomes. While this is a useful way to store DNA, it also makes it inaccessible to many proteins and other molecules that activate genes, copy DNA or perform other important cell processes. To enable these processes to take place, the cell can selectively disassemble particular nucleosomes and remove the histone proteins. Afterwards, the nucleosomes must reassemble to repackage the DNA.

A single nucleosome contains four pairs of histones, with two pairs consisting of a H3 and a H4 histone. Histone chaperones assemble nucleosomes in a two-step process. First, two of these histone H3-H4 pairs (collectively known as a tetramer) interact with DNA to form a group or "complex". Then, two more pairs of different histones bind to complete the nucleosome. An enzyme called CAF1 is known to attach H3-H4 tetramers onto DNA as the DNA is being copied, which allows nucleosomes to form on the newly made DNA. However, it is not known how CAF1 deposits H3-H4 tetramers onto the DNA.

Sauer et al. explored how yeast CAF1 works by carrying out a series of experiments in a cell-free system. The experiments showed that each CAF1 enzyme binds to a single H3-H4 pair. When attached to their histone cargo, two CAF1 enzymes bind to DNA and attach a H3-H4 tetramer onto it. The tetramer has to form in this way for histones to be correctly delivered to DNA after the DNA has been copied.

Sauer et al. also identified a new region of the CAF1 enzyme that binds to DNA. Together with another region, this enables CAF1 to bind to an extended stretch of DNA that accommodates the H3-H4 tetramer. Together, the findings explain the sequence of events that take place when CAF1 attaches H3-H4 tetramers onto DNA in the first step of nucleosome formation. Future work will be required to understand the structure of CAF1 in different situations and to find out how the cell targets this enzyme to stretches of DNA that have just been copied.

suggested that the parental histone modifications are propagated to newly incorporated nucleosomes upon cell division (reviewed in *Margueron and Reinberg, 2010*; *Probst et al., 2009*), however whether a purely histone-based inheritance mechanism exists remains a matter of debate (*Ptashne, 2013*), and the mechanism of how histone modifications are maintained following DNA replication remains unclear.

Cumulative evidence supports the model that upon replication, parental H3-H4 are conservatively propagated as tetramers in proliferating cultured cells (*Jackson, 1988*; *Prior et al., 1980*). Newly deposited H3-H4 tetramers are assembled entirely of new histones thus precluding the acquisition of new histone modifications based on preexisting parental modifications within the same nucleosome (*Katan-Khaykovich and Struhl, 2011*; *Xu et al., 2010*). Thus, one mechanism to ensure the maintenance of epigenetic modification could involve a read-write mechanism in which parental histone modifications are copied over to newly incorporated nucleosomes (*Probst et al., 2009*; *Ragunathan et al., 2015*).

A central, unanswered question is how parental H3-H4 tetramers are propagated and how new H3-H4 tetramers are assembled behind the replication fork. The conservative nature of tetramer propagation and de novo assembly could be due to the biochemical properties of the histone chaperone machinery that operates at the replication fork. A number of histone chaperones including antisilencing function 1 (Asf1) and DAXX bind H3-H4 dimers but not tetramers, indicating that chaperon-mediated tetramer assembly requires a two-step mechanism (*Elsässer et al., 2012*; *English et al., 2006*; *Liu et al., 2012a*; *Natsume et al., 2007*). Asf1 associates with H3-H4 and Mcm2, a subunit of the replicative helicase (*Groth et al., 2007*). A crystal structure of the complex and binding studies reveal a 1:1:1:1 stoichiometry indicating that dimeric H3-H4 is propagated at replication forks (*Huang et al., 2015*; *Richet et al., 2015*). Thus, a current model suggests that the Mcm2-Asf1 complex mediates the passing of parental H3-H4 through transient tetramer disruption and conservative reassembly onto DNA behind the replication fork (*Clément and Almouzni, 2015*; *Huang et al., 2015*; *Richet et al., 2015*). This model underlines the importance of H3-H4 dimers as

intermediates even for the recycling of parental histones and raises the question of how such dimers are reassembled into tetramers behind the replication fork.

Chromatin assembly factor 1 (CAF1) is a histone chaperone complex that deposits new H3-H4 de novo in a DNA-synthesis-dependent manner (*Smith and Stillman, 1989*) and is functionally conserved throughout eukaryotes. CAF1 contains three subunits: p150, p60 and p48 (Cac1, Cac2 and Cac3 in *Saccharomyces cerevisiae*) (*Kaufman et al., 1997*; *Verreault et al., 1996*). CAF1 is recruited to the replication fork by interaction of the p150 subunit with proliferating cell nuclear antigen (PCNA), the DNA polymerase processivity clamp (*Gérard et al., 2006*; *Shibahara and Stillman, 1999*). Human CAF1 preferentially interacts with the replication-dependent histones H3.1/2 but not with the replication-independent histone variant H3.3 (*Benson et al., 2006*; *Tagami et al., 2004*). Budding yeast contains only a H3.3 ortholog which is used in both replication-dependent and independent H3 deposition pathways.

The C-terminal region of p150 contains a conserved Winged-helix DNA-binding domain (WHD) that is thought to contribute to stabilization of CAF1 at the replication fork (*Zhang et al., 2016*). The p60 subunit preferentially associates with Asf1b in vivo (*Abascal et al., 2013*; *Gurard-Levin et al., 2014*; *Tagami et al., 2004*), and Asf1 is thought to deliver H3-H4 dimers to CAF1 for deposition onto DNA (*Tyler et al., 1999*). The p48 subunit (Rb-associated protein RbAp48) interacts with a single H3-H4 dimer through the N-terminal tail of H3 and the N-terminal helix of H4 (*Nowak et al., 2011*; *Schmitges et al., 2011*; *Song et al., 2008*; *Zhang et al., 2013*). However, in the context of CAF1, the N-terminal H3 and H4 tails are not essential for histone deposition activity, indicating that additional contacts are made between p150 and/or p60 and the H3-H4 core histone fold (*Shibahara et al., 2000*; *Winkler et al., 2012*), a model supported by recent hydrogen/deuterium exchange (HX) data and cross-linking MS (*Kim et al., 2016*; *Liu et al., 2016*). Whether CAF1 binds a H3-H4 dimer or tetramer has remained controversial. Human CAF1 has been found to bind a single H3-H4 dimer (*Benson et al., 2006*; *Tagami et al., 2004*), whereas yeast CAF1 (yCAF1) has been reported to bind H3-H4 tetramers in a non-canonical conformation in vitro and in vivo and prior to deposition onto DNA (*Liu et al., 2012b*; *Winkler et al., 2012*). Further, it has been proposed that the Cac1 subunit in isolation is sufficient to enable H3-H4 tetramerization (*Liu et al., 2016*). Together, the model is emerging that H3-H4 are maintained as dimers from synthesis up until tetramer assembly by CAF1. Whether the tetramerization of H3-H4 occurs on CAF1 prior to deposition onto DNA or via a sequential CAF1-mediated deposition of two H3-H4 dimers onto DNA, remains unclear.

To explore the molecular mechanism underlying H3-H4 deposition by CAF1, we performed structure-function analysis of yCAF1 and of the yCAF1-H3-H4 complex and analyzed the histone deposition reaction onto DNA. We report that yCAF1 binds a single H3-H4 heterodimer and prevents H3-H4 tetramer formation. Our data imply that two yCAF1-H3-H4 complexes cooperate for assembly and deposition of H3-H4 tetramers. Biochemical studies show that the Cac1 subunit contacts DNA through a DNA-binding domain that is located in the region comprising the predicted coiled-coil segment of Cac1 and that high affinity DNA-binding also requires the WHD domain. High affinity DNA-binding by yCAF1 requires a B-form DNA substrate in the range of ~40–80 bp, due to cooperative binding of two yCAF1 complexes. Such extended DNA substrates allow deposition of H3-H4 tetramers by yCAF1. While DNA binding does not prove necessary for tetramer deposition using purified components, it is required for DNA synthesis-coupled nucleosome assembly in an in vitro assembly system. DNA-binding deficient mutants retained the ability to bind H3-H4 heterodimers yet histone binding per se was not sufficient for the chaperone activity of yCAF1. In addition, H3-H4 binding to yCAF1 and yAsf1 or Mcm2 was mutually exclusive suggesting a possible hand-over mechanism for final H3-H4 tetramer deposition onto DNA by yCAF1. Finally, we report that H3-H4 tetramerization is required for release of H3-H4 from yCAF1 during DNA deposition. We thus propose a model in which two yCAF1-H3-H4 complexes cooperatively bind to an extended DNA element enabling deposition of two copies of H3-H4, the first step in nucleosome formation.

## Results

### Expression and purification of yCAF1 variants

Budding yeast CAF1 (yCAF1) is a heterotrimeric complex containing the Cac1, Cac2 and Cac3 subunits (*Kaufman et al., 1997*). Cac1 contains a predicted coiled-coil region rich in amino acid residues K/E/R, an acidic E/D domain and a C-terminal WHD, which was shown to interact with DNA (*Zhang et al., 2016*). Cac2 and Cac3 are predicted WD40 domain proteins (*Figure 1A*). While Cac2 contains a C-terminal Asf1 interaction motif, Cac3 is predicted, based on sequence similarity to the mammalian ortholog RbAp48, to interact with histones H3-H4. To determine the molecular architecture of the yCAF1 complex, we established a co-expression system for Cac1, Cac2 and Cac3 in insect cells (*Figure 1A*). Expression and purification of this complex yielded a stable, monodisperse heterotrimeric yCAF1 complex with 1:1:1 stoichiometry, as judged from size-exclusion chromatography coupled to multi-angle laser light scattering (SEC-MALLS; *Figure 1B*, *Table 1*). This complex was able to bind histones H3-H4 to yield a homogeneous complex that, despite the additional mass, had a slightly smaller hydrodynamic radius as indicated by mobility on the size-exclusion column (see also below). To derive further insights into the architecture of the yCAF1 complex and its interaction with H3-H4, we used limited proteolysis to identify stable yCAF1 subcomplexes. In agreement with disorder predictions (*Figure 1—figure supplement 1A*), we found that while Cac2 and Cac3 were mostly protease cleavage resistant, Cac1 was readily cleaved into smaller fragments (*Figure 1—figure supplement 1B*). Mass spectrometry analysis revealed a C-terminal fragment of Cac1 (Cac1T; amino acid residues 230–606). Using this information, we designed a series of N- and C-terminal Cac1 truncations (*Figure 1A*) and co-expressed these with Cac2 and Cac3 in insect cells. We found that all the Cac1 truncation variants were able to bind to Cac2 and Cac3 allowing purification of stable and monodisperse heterotrimeric complexes (*Figure 1C,D*). These data show that the Cac1 region required for interaction is located in the segment spanning amino acid residues 230–494. All the yCAF1 variants described here retained the ability to bind H3-H4, indicating that the deleted regions of Cac1 are not essential for histone interaction. Additional limited proteolysis revealed yCAF1 complex dissociation during SEC analysis (*Figure 1—figure supplement 1C*). LC-MS analysis showed that peak two contained a complex of Cac1 comprising amino acid residues 234–442 and close to full-length Cac3 spanning amino acid residues 5–422. This complex did not retain Cac2 binding activity, which migrated as separate peak on the SEC column (*Figure 1—figure supplement 1C*). This Cac2 fragment, spanning residues 1–434, was only missing the B domain, a known binding site for yAsf1 (*Malay et al., 2008*). We therefore tentatively assign the Cac2 and Cac3 binding regions on Cac1 to amino acid residues 443–489 and 234–442, respectively. The assignment of the Cac2 and Cac3 binding regions on Cac1 is in agreement with earlier data from yeast and human CAF1 and emphasizes the conserved nature of the CAF1 complex (*Kaufman et al., 1995*; *Krawitz et al., 2002*).

### yCAF1 binds a single H3-H4 heterodimer and prevents tetramerization

The yCAF1 complex is thought to enable H3-H4 tetramerization before final deposition onto DNA (*Liu et al., 2012b*, *2016*; *Winkler et al., 2012*). However, previous immunoprecipitation experiments show that human CAF1 binds a single copy of H3-H4 (*Tagami et al., 2004*). To resolve this apparent controversy, we performed a series of biophysical measurements including SEC-MALLS, analytical ultracentrifugation (AUC) and native mass spectrometry (native MS), a technique with high mass accuracy. SEC-MALLS showed a mass of 172 kDa for yCAF1, consistent with the expected mass of a heterotrimer (*Figure 1B*, *Table 1*). In the presence of H3-H4 we found single species with a mass of 199 kDa, in agreement with binding of one yCAF1 trimer bound to one H3-H4 dimer. We obtained the same mass even when H3-H4 was added in twofold molar excess. All four mutant yCAF1 complexes analyzed also formed monodisperse heterotrimers that bound a single copy of H3-H4 (*Figure 1C,D* and *Table 2*).

Native MS of yCAF1 showed an ion population at high *m/z* corresponding to a mass of 174 kDa, in agreement with the mass of intact yCAF1 (*Figure 1E*; *Table 3*). At low *m/z* we observed peaks of free Cac2 (53 kDa) and Cac3, (50 kDa) while the intermediate *m/z* range showed masses corresponding to Cac1-Cac3 and Cac2-Cac3 subcomplexes. In presence of histones, yCAF1 was bound to a single copy of H3-H4 (201 kDa) (*Figure 1F*). The 32$^+$ ion population of yCAF1-H3-H4 was subjected to

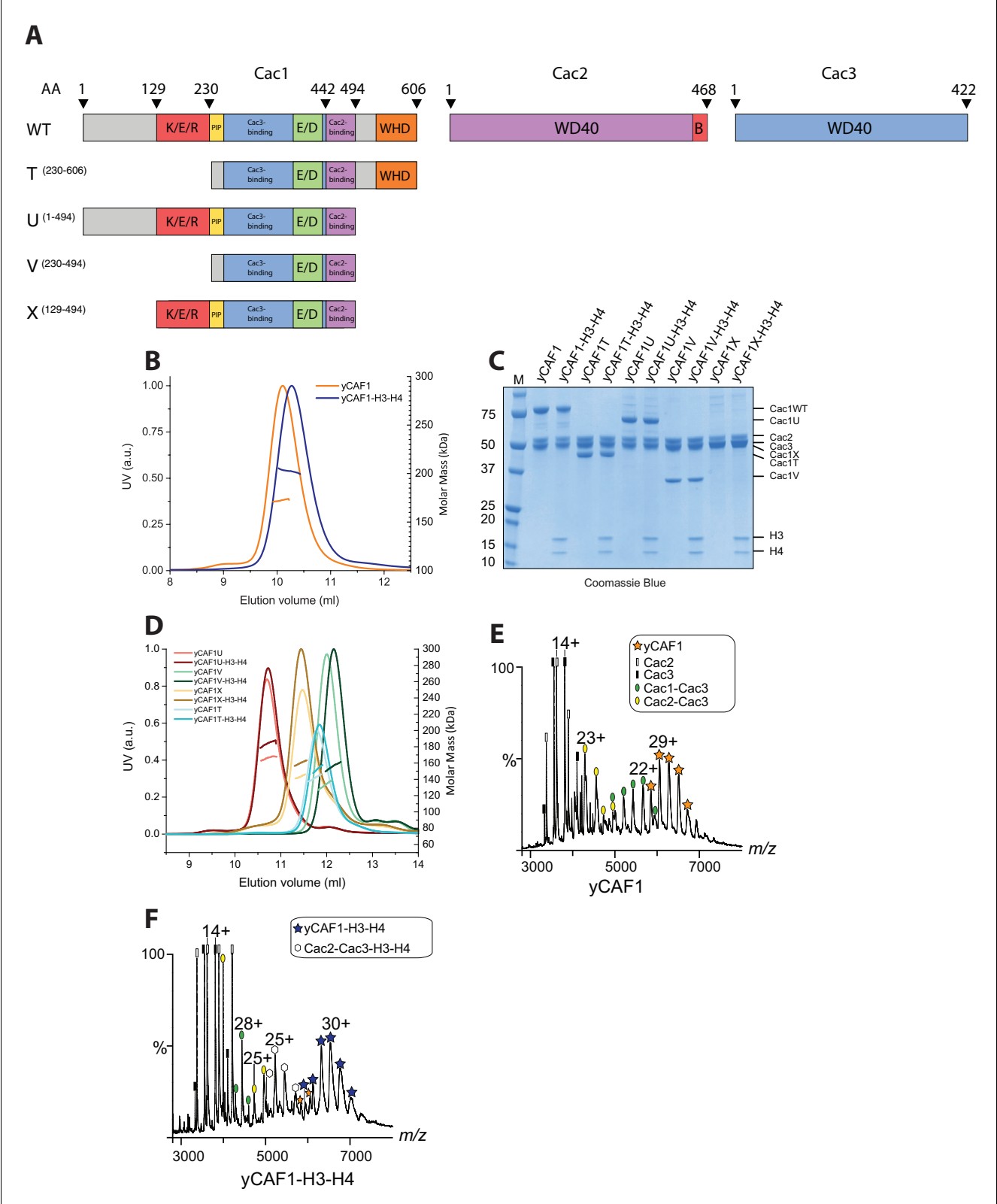

**Figure 1.** Domain architecture and purification of yCAF1. (**A**) Domain architecture of Cac1, Cac2 and Cac3 with K/E/R domain containing a predicted coiled-coil segment; PIP, PCNA interacting peptide; E/D, acidic domain; WHD, winged helix domain; WD40-repeat β-propeller domain; B, Asf1 interaction domain. Constructs used are shown below. yCAF1T and yCAF1V are missing the first two residues of the PIP motif (227-Q-x-x-I-x-x-F-F-234) in Cac1 (**B**) Determination of the apparent molecular mass of yCAF1 ± H3-H4 using SEC-MALLS. Lines correspond to the UV$_{280nm}$ traces of yCAF1 (red)

*Figure 1 continued on next page*

*Figure 1 continued*

or yCAF1–H3-H4 (blue) eluting from the SEC column. Dots correspond to the molar mass determined for yCAF1 or yCAF1 –H3-H4. (C) SDS-PAGE analysis of purified yCAF1 constructs ± H3-H4. (D) Determination of the apparent molecular mass of yCAF1 mutants ± H3-H4 using SEC-MALLS. The $UV_{280nm}$ traces are shown as lines and the molar mass measurements as dots. (E) Native mass spectra of yCAF1 in (E) the absence or (F) the presence of H3-H4. The yCAF1 (*H*), yCAF1-H3-H4 (*H*) and the subcomplexes are labeled.

The following figure supplements are available for figure 1:

**Figure supplement 1.** Structural and biochemical characterization of yCAF1 complexes.

**Figure supplement 2.** MS analysis of yCAF1-H3-H4 and H3-H4.

tandem MS experiments and confirmed a 1:1:1:1:1 stoichiometry of the yCAF1-H3-H4 complex (*Figure 1—figure supplement 2A*). Thus, consistent with previous results, we found that isolated yCAF1 forms a heterotrimer containing a single copy of each subunit (*Liu et al., 2012b*; *Winkler et al., 2012*). However, none of our data support the model that yCAF1 binds two H3-H4 heterodimers. To further test this, we added excess H3-H4 to yCAF1 and analyzed the sample by native MS. The largest species obtained corresponded to yCAF1 bound to a single H3-H4 heterodimer (*Figure 1—figure supplement 2B*). Of note, the masses observed for excess H3-H4 correspond to those of tetramers (*Figure 1—figure supplement 2B*). Native MS analysis of the H3-H4 sample also showed masses corresponding to tetramers indicating that H3-H4 remain associated as tetramers in this experiment (*Figure 1—figure supplement 2C*, *Table 3*). Considering that isolated H3-H4 forms tetramers in solution at equivalent concentrations (*Winkler et al., 2012*) and in our native MS experiments (*Figure 1—figure supplement 2C*, *Table 3*), our data suggest that yCAF1 prevents H3-H4 tetramerization.

**Table 1.** Overall biophysical parameters of yCAF1. Column labeling: SEC-MALLS (Size-exclusion chromatography - multi-angle laser light scattering); EQ-AUC (equilibrium analytical ultracentrifugation); SV-AUC (sedimentation velocity analytical ultracentrifugation); SAXS (small angle X-ray scattering); Native MS (native mass spectrometry); $MM_{SLS}$ (Molar masses determined by SEC-MALLS); $MM_{AUC}$ (Molar masses determined by equilibrium analytical ultracentrifugation); $s^0_{20,w}$ (Sedimentation coefficient determined by velocity analytical ultracentrifugation); $s_{th}$ (computed sedimentation coefficient derived from SAXS envelopes); $R_g$ (radius of gyration); $MM_{SAXS}$ (molar masses determined by SAXS); $D_{max}$ (maximum dimension); $V_p$ (excluded particle Volume); $MM_{MS}$ (molar masses determined by native MS; $MM_{th}$ (theoretical molar mass calculated).

The errors reported for SEC-MALLS are the residual standard deviations of the observed data from the fitted values calculated using Astra. The errors of the AUC experiments are derived from the standard deviations of linear fits of the obtained data points to extrapolate the respective values ($MM_{AUC}$ and $s^0_{20,w}$) to zero protein concentration. The errors reported for the parameters derived from SAXS are based on the observed range of results is it possible to obtain, adjusting (within acceptable theoretical limits) the data points used for the calculation and as such represent the confidence range of the parameter. Resolution of the *ab inito* SAXS models was calculated according to (*Tuukkanen et al., 2016*). Errors in native MS were determined according to (*McKay et al., 2006*). $C_{SLS}$, $C_{SAXS}$, $C_{MS}$ are the concentrations of samples used for SEC-MALLS, SAXS and native MS respectively. N.D. (not determined)

| | Sec-malls | Eq-auc | | Sv-auc | SAXS | SAXS | SAXS | SAXS | SAXS | Native MS | |
|---|---|---|---|---|---|---|---|---|---|---|---|
| Sample | $MM_{SLS}$ kDa ($C_{SLS}$, μM) | $MM_{AUC}$ kDa | $s_{th}$ Svedberg | $s^0_{20,w}$ Svedberg | $R_g$ nm ($C_{SAXS}$ mg·ml$^{-1}$) | $MM_{SAXS}$ kDa | $D_{max}$ nm | $V_p$ nm$^3$ | Resolution Å | $MM_{MS}$ kDa ($C_{MS}$ μM) | $MM_{th}$ kDa |
| yCAF1 | 172.4 ± 1% (67) | 180 ± 10 | 6.2 | 6.41 ± 0.03 | 6.39 ± 0.22 (11.4) | 175 ± 3 | 26 ± 2 | 307 ± 5 | 57 ± 4 | 174.49 ± 0.30 (2.5) | 174.0 |
| yCAF1-H3-H4 | 198.9 ± 1.1% (50) | 200 ± 11 | 7.1 | 6.84 ± 0.06 | 6.02 ± 0.35 (10) | 203 ± 13 | 25 ± 2 | 355 ± 23 | 54 ± 4 | 201.00 ± 0.01 (7) | 200.7 |
| yCAF1T | 142 ± 1% (15) | N.D. | N.D. | N.D. | 5.66 ± 0.03 (30.0) | 127 ± 1 | 20 ± 1.2 | 222 ± 2 | 49 ± 4 | N.D. | 146.8 |
| yCAF1T-H3-H4 | 153 ± 1% (15) | N.D. | N.D. | N.D. | 5.10 ± 0.03 (13.2) | 163 ± 1 | 17.3 ± 1.3 | 285 ± 2 | 52 ± 4 | N.D. | 173.6 |

**Table 2.** Summary of SEC-MALLS data for yCAF1 variants. Column labeling: SEC-MALLS (Size-exclusion chromatography - multi-angle laser light scattering); $MM_{SLS}$ (Molar masses determined by SEC-MALLS); $MM_{th}$ (theoretical molar mass calculated). $C_{SLS}$ is the concentration used for SEC-MALLS. Errors reported are the residual standard deviations of the observed data from the fitted values calculated using Astra.

| Sample | $MM_{SLS}$ kDa ($C_{SLS}$, μM) | $MM_{th}$ kDa |
|---|---|---|
| yCAF1U | 166 ± 1.1% (20) | 161.4 |
| yCAF1U-H3-H4 | 184 ± 1% (20) | 188.2 |
| yCAF1V | 133 ± 1.2% (20) | 134.4 |
| yCAF1V-H3-H4 | 156 ± 1.1% (20) | 161.2 |
| yCAF1X | 143 ± 1% (15) | 147.2 |
| yCAF1X-H3-H4 | 160 ± 1% (15) | 174.0 |

In addition, we carried out AUC sedimentation velocity experiments on yCAF1 and yCAF1-H3-H4 (*Figure 2A,B*). A *g(s\*)* analysis showed that there was a minor shift in the peak position with an increase in the concentration yCAF1 or yCAF1-H3-H4, indicating that predominantly a single species was present at all examined concentrations. The sedimentation coefficients are in agreement with a monomeric complex in solution and also with theoretical Svedberg values calculated from SAXS envelopes (see below). Sedimentation equilibrium experiments showed that the samples contained a single monodisperse species and the residuals showed mostly random distribution (*Figure 2—figure supplement 1*). For yCAF1, the data yielded a molecular mass of 179 kDa, and for yCAF1-H3-H4 of 199 kDa, values close to the calculated masses of these complexes considering equal stoichiometry of the polypeptide chains (*Table 1*).

**Table 3.** Summary of native mass spectrometry.

| Protein sample | Concentration (μM) | Oligomerization state | Measured mass ± error (Da)* | Calculated mass (Da) |
|---|---|---|---|---|
| yCAF1 | 2.5 | Cac1:Cac2:Cac3 | 174 492 ± 3 | 173965.1 |
| yCAF1 | 2.5 | Cac1:Cac3 | 123 927 ± 5 | 120735 |
| yCAF1 | 2.5 | Cac2:Cac3 | 103 840 ± 4 | 103755.1 |
| yCAF1 | 2.5 | Cac2 | 53 273 ± 6 | 53230.1 |
| yCAF1 | 2.5 | Cac3 | 50 568 ± 7 | 50525 |
| yCAF1-H3-H4 | 7 | Cac1:Cac2:Cac3:H3:H4 | 201 002 ± 5 | 200 720.3 |
| yCAF1-H3-H4 | 7 | Cac2:Cac3:H3:H4 | 130 343 ± 7 | 130510.4 |
| H3-H4 | 10[†] | (H3-H4)$_2$ | 53 015 ± 4 | 53510.6 |
| H3-H4 | 10[†] | H3-H4 | 26 508 ± 2 | 26755.3 |
| H3-H4 | 10[†] | H3 | 15 271 ± 2 | 15388 |
| H3-H4 | 10[†] | H4 | 11236 ± 3 | 11367.3 |

*Values reported represent the mean value ± standard deviation according to (*McKay et al., 2006*). Combinations of neighboring *m/z* values were used to determine distinct M values of a macromolecule. Using these values, a mean value of M and its standard deviation were calculated.
[†]Values reported assume that H3-H4 are tetrameric in solution.

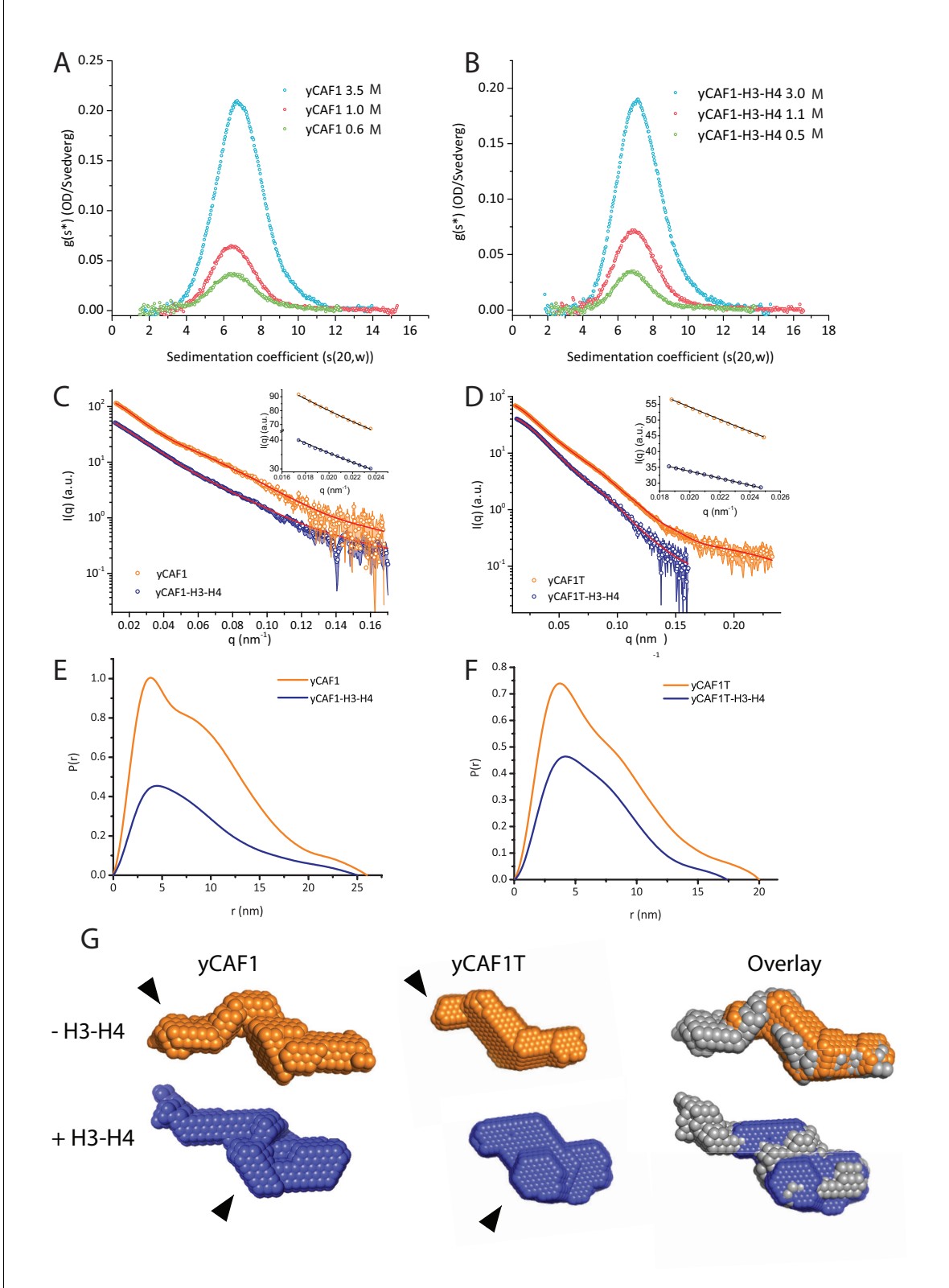

**Figure 2.** yCAF1 binds a single H3-H4 heterodimer. (**A**) Sedimentation velocity analytical ultracentrifugation of the yCAF1 complex. (**B**) yCAF1-H3-H4. Shown is the plot of the sedimentation coefficient distribution at different protein concentrations. (**C**) The experimental SAXS profile (log intensity (**I**) as a function of the momentum transfer (**q**)). Dots with error bars are the experimental scattering data. yCAF1 ± H3-H4 (blue or orange respectively). The normalized fit to the experimental data is superimposed as a black line. Inset: Guinier plot (log I vs. (**q**$^2$) of the low q region of the X-ray scattering data

*Figure 2 continued on next page*

*Figure 2 continued*

(D) yCAF1T ± H3-H4 (blue or orange, respectively). Bottom panels: Normalized interatomic distance distribution functions. (E) The *p(r)* distribution plot for yCAF1 alone (blue) and bound to H3-H4 (orange). (F) The *p(r)* distribution plot for yCAF1T alone (blue) and bound to H3-H4 (orange). (G) Average DAMMIN bead models. Left: yCAF1 (orange) and yCAF1-H3-H4 (blue). Middle: yCAF1T (orange) and yCAF1T-H3-H4 (blue). Right: Superposition of yCAF1 (grey) onto yCAF1T (orange) and yCAF1-H3-H4 (grey) onto yCAF1T-H3-H4 (blue). Arrows indicate the N-terminal extension of Cac1 and the possible positioning of H3-H4.

The following figure supplement is available for figure 2:

**Figure supplement 1.** Analytical ultracentrifugation of yCAF1 complexes.

To further analyze the structure of yCAF1, we performed small angle X-ray scattering measurements of yCAF1 and yCAF1T complexes unbound or bound to H3-H4. yCAF1T lacks the N-terminal region of the Cac1 subunit (*Figure 1A*). To reduce potentially confounding interparticle effects, we measured scattering data using an in-line size exclusion chromatography system (SEC-SAXS) (*Pernot et al., 2013*). The scattering curves showed no sample aggregation, and the linear Guinier range indicated monodisperse protein complexes for all four complexes analyzed (*Figure 2C,D* and insets). From these data, we obtained an $R_g$ of 6.39 nm for yCAF1 while the distance distribution function *p(r)* showed skewed shape characteristic of elongated, multi-domain particles, with a maximum diameter ($D_{max}$) of 26 nm (*Figure 2E*). For yCAF1-H3-H4, we obtained an $R_g$ of 6.02 and a $D_{max}$ of 25 nm (*Figure 2E*, *Table 1*). The molecular masses determined from SAXS were 175 kDa (yCAF1) and 203 kDa (yCAF1-H3-H4), in close agreement with those determined from orthogonal methods (*Table 1*). *Ab initio* reconstructions showed an elongated particle for yCAF1 (*Figure 2G*). In the H3-H4-bound form, a similar elongated shape was obtained with additional mass towards the center of the particle (*Figure 2G*). *Ab initio* calculation of the Svedberg coefficients of yCAF1 and yCAF1-H3-H4 using the molecular envelopes obtained from the SAXS experiments also match the measured Svedberg values from sedimentation velocity AUC experiments (*Table 1*), also emphasizing the elongated shape of the particle. For yCAF1T we obtained an $R_g$ of 5.66 nm and a $D_{max}$ of 20 nm and for the H3-H4-bound complex an $R_g$ of 5.10 nm and a $D_{max}$ of 17.3 nm (*Figure 1D,F* and *Table 1*). *Ab initio* reconstructions also showed an elongated particle for yCAF1T (*Figure 2G*). As for yCAF1, addition of H3-H4 to yCAF1T, resulted in additional mass towards the center of the particle. Based on the overall reduction of the particle diameter of yCAF1T, which matches the expected volume of the flexible N-terminal region of Cac1, we suggest that the bulk of this segment is located there. Comparison of H3-H4-bound and unbound yCAF1 variants indicates that the histone heterodimer is positioned towards the center of the particle (*Figure 2G*). We consistently observed, as also shown in SEC-MALLS (*Figure 1B*), a smaller radius of gyration for the H3-H4-bound yCAF1 complexes, indicative of conformational compaction upon histone binding. Together, a series of rigorous and complementary biophysical techniques led us to conclude that yCAF1 forms an elongated trimer with 1:1:1 stoichiometry, which binds a single H3-H4 heterodimer.

## DNA-binding elements of the yCAF1 complex

Previous data show that Cac1 contains a C-terminal WHD DNA-binding domain (*Zhang et al., 2016*). The WHD binds to a 10–16 bp DNA element in a non-specific fashion with a dissociation constant $K_d$ of ~2 μM (*Zhang et al., 2016*). As previous studies were limited to the isolated WHD, we investigated the DNA-binding requirements of full-length yCAF1. Purification of the yCAF1 complex from insect cells revealed the presence of contaminating nucleic acids and insect cell histones H2A, H2B, H3 and H4 suggesting that yCAF1 potentially directly interacts with nucleosomes (*Figure 3—figure supplement 1A*). To test this model, we analyzed binding of yCAF1 to reconstituted nucleosome core particles using recombinantly produced *Xenopus* histones, which were assembled on a 147 bp DNA fragment containing the 601 nucleosome positioning sequence (*Lowary and Widom, 1998*). Native PAGE analysis showed that these preparations contained nucleosomes and a fraction of unbound 147 bp DNA (*Figure 3—figure supplement 1B*). Titration of the yCAF1 complex showed that in this direct competition assay, yCAF1 preferentially interacted with the naked DNA but not with the nucleosome core particles (*Figure 3—figure supplement 1B*, lanes 2–8). Thus,

surfaces of H3-H4 that are required for yCAF1 binding are buried once these histones are assembled into nucleosomes.

We found that yCAF1 bound to the 147 bp DNA fragment with a dissociation constant $K_D$ of ~2.1 µM (*Figure 3A*, *Table 4*). Quantitative analysis of the binding reaction showed a Hill coefficient of 2.0, indicative of cooperative binding. yCAF1U, a construct lacking the WHD domain had a $K_D$ of ~4.1 µM, (*Figure 3A*, *Table 4*). The non-conserved N-terminal region (amino acids 1–129) did not contribute significantly to DNA binding as a construct lacking this segment (CAF1X) had similar DNA-binding affinity (5.7 µM). yCAF1U and yCAF1X showed less steep binding isotherms but still largely retained their ability to assemble cooperatively as judged by a positive Hill coefficient (*Table 4*). yCAF1T, a variant lacking the N-terminal 229 amino acids including the K/E/R-rich coiled-coil, showed no detectable DNA binding, despite the fact that it contained the WHD domain indicating that the DNA-binding activity of the WHD domain is masked. yCAF1V, which contained a deletion of the WHD, showed also no detectable DNA binding (*Figure 3A*). Taken together, these data indicate that the region spanning the K/E/R-rich coiled-coil segment of Cac1 contains a DNA-binding domain which appears to hierarchically cooperate with the C-terminal WHD for high-affinity DNA binding as constructs lacking this domain are devoid of DNA binding. The WHD contributes to overall DNA-binding avidity but is not sufficient to enable high affinity binding in the absence of the coiled-coil domain.

Analysis of the DNA sequence length preferences, using a DNA ladder ranging from 20 to 100 bp DNA size (*Table 5*), showed that yCAF1 bound DNA fragments longer than >40 bp more efficiently than shorter ones (*Figure 3B,C*). While this would not be unusual for a DNA-binding factor, the fact that yCAF1 discriminated against DNA shorter than 40 bp was unexpected as the WHD binds DNA of 10–16 bp length (*Zhang et al., 2016*). To further test the interplay between DNA substrate length and yCAF1 binding efficacy we investigated whether yCAF1 can also cooperatively bind to DNA molecules that are shorter than 40 bp in length. We performed DNA-binding assays with 17 bp, 42 bp or 84 bp DNA substrates and analyzed the binding isotherms (*Figure 3C*). In agreement with our previous results we observed that while 42 bp and 84 bp DNA were clearly bound cooperatively by yCAF1 (Hill coefficient of 2.2 for both substrates), the 17 bp DNA substrate displayed standard Michaelis-Menten kinetics (Hill coefficient of 1.3) indicative of a single binding site (*Table 4*). Of note, while yCAF1 efficiently bound to free B-DNA, the DNA geometry of the nucleosome core particle, a reaction product of yCAF1-mediated assembly, is not compatible with yCAF1 interaction (*Figure 3—figure supplement 1B*). Together, our results suggest that yCAF1 uses a two-pronged DNA-binding mode involving the WHD and the coiled-coil segment of Cac1. Optimal yCAF1 DNA binding requires regular B-DNA geometry and a minimum of ~40 bp length.

## DNA-binding by yCAF1 is not essential for tetrasome deposition with purified components

Considering that two yCAF1-H3-H4 complexes need to come together for assembly of two copies of H3-H4 on DNA to form so-called 'tetrasomes', we hypothesized that the DNA length requirements are due to binding of two complexes to an extended DNA substrate for histone deposition. Parenthetically, a DNA substrate of similar length (~60–80 bp) is also required for tetramer binding in the nucleosome (*Luger et al., 1997*).

To assess the ability of yCAF1 to assemble tetrasomes, we incubated yCAF1-H3-H4 with a 84 bp DNA fragment derived from the H3-H4 binding region of the 601 nucleosome positioning sequence (*Lowary and Widom, 1998*) and of sufficient length for salt deposition of a single tetramer (*Figure 4—figure supplement 2A*). The reactions were analyzed on native PAGE and stained for DNA (SYBR Safe, left panel) or protein (Coomassie, right panel). As expected, salt-deposition showed H3-H4 tetrasomes (*Figure 4A*, lane 2 and *Figure 4—figure supplement 2A*). A histone H3 L126R/I130R mutant (H3M), which disrupts tetramer formation (*Winkler et al., 2012*), showed a band with higher mobility, interpreted to represent disomes, a dimer of H3M-H4 bound to DNA (*Figure 4A*, lane 1 and *Figure 4—figure supplement 2B*). Titration of the yCAF1-H3-H4 complex onto DNA showed the appearance of tetrasomes but no apparent disome assembly intermediates (*Figure 4A*, left panel, lanes 3–10). The top part of the gel also showed yCAF1 bound to DNA, apparently in the absence of H3-H4 (see below).

The yCAF1T, yCAF1U, yCAF1V and yCAF1X complexes bound to H3-H4 also retained to varying degrees tetrasome deposition activity (*Figure 4B*, *Figure 4—figure supplement 1A–C*). yCAF1U,

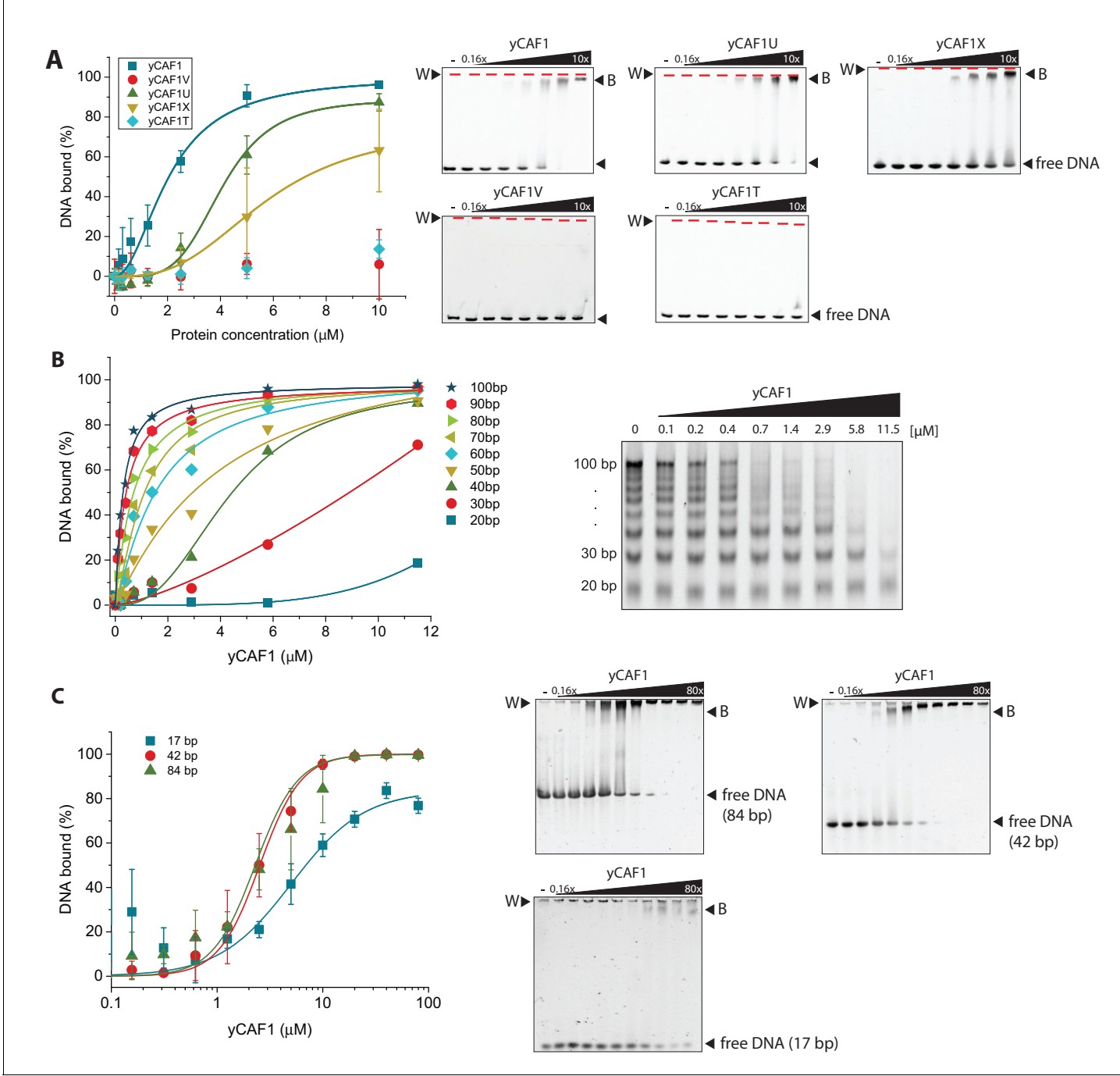

**Figure 3.** yCAF1 binding to DNA. (**A**) Left panel: Binding curves of yCAF1 variants to 147 bp DNA. Right panels: EMSA showing binding of yCAF1 variants to 147 bp DNA. Free DNA and yCAF1-bound (**B**) DNA are indicated. Wells (W) are indicated additionally with red horizontal bars. Increasing amounts of yCAF1 (0.15, 0.3, 0.61, 1.25, 2.5, 5 or 10 µM) were mixed with 1 µM DNA. Error bars represent SEM values of three technical replicates (**B**) Left panel: binding curves of yCAF1 binding to DNA fragments of 20–100 bp (Sequence information in **Table 5**). Right panel: EMSA showing yCAF1 binding to free DNA fragments of 20–100 bp. Concentration of the DNA 10–100 bp ladder was 275 nM overall nucleotide base pairs present in the binding reaction. yCAF1- DNA binding was quantified by measuring DNA substrate depletion. (**C**) Left panel: Binding curves of yCAF1 to 17 bp, 42 bp and 84 bp DNA. Right panels: EMSA showing binding of yCAF1 to 17 bp, 42 bp and 84 bp DNA. Wells (W), free DNA and yCAF1-bound (**B**) DNA are indicated. Increasing amounts of yCAF1 (0.15, 0.3, 0.61, 1.25, 2.5, 5, 10, 20, 40 or 80 µM) were mixed with 1 µM DNA. Error bars represent SEM values of three technical replicates.

The following figure supplement is available for figure 3:

*Figure 3 continued on next page*

*Figure 3 continued*

**Figure supplement 1.** Analysis of nucleosome binding by yCAF1.

which lacks the WHD, showed normal tetrasome deposition (*Figure 4—figure supplement 1B*). yCAF1X, which has an additional deletion of amino acids 1–128 of Cac1, also showed wild-type levels of tetrasome deposition (*Figure 4—figure supplement 1C*). The DNA-binding deficient yCAF1T deposited tetrasomes and a complex with slightly higher mobility, presumably a tetrasome in a non-canonical position on the DNA substrate (*Figure 4—figure supplement 1A*). yCAF1V also showed tetrasome assembly activity and in addition a disome assembly intermediate (*Figure 4B*, lanes 6–8).

In all cases, addition of DNA resulted in release of H3-H4 from yCAF1 as seen by the appearance of tetrasomes and free yCAF1. This was most clearly seen with the DNA-binding deficient variants yCAF1T and yCAF1V (*Figure 4B* and *Figure 4—figure supplement 1A*). With both mutant complexes, addition of yCAF1-H3-H4 to DNA resulted in the appearance of tetrasomes and free yCAF1T or yCAF1V. Only when the free DNA was exhausted from the reaction, the excess yCAF1T-H3-H4 or yCAF1V-H3-H4 complexes were seen (*Figure 4B* and *Figure 4—figure supplement 1A*, right panels, lanes 8–10). As expected for these DNA-binding deficient variants, free yCAF1T or yCAF1V, liberated during the histone deposition reaction, did not interact with the DNA (*Figure 4B*, *Figure 4—figure supplement 1A*, left panels, lanes 3–10). For the DNA-binding competent yCAF1U and yCAF1X, as for wild-type yCAF1, we observed in addition to tetrasome deposition, that the released yCAF1U or yCAF1X further partitioned into a DNA-bound complex at lower concentrations (*Figure 4—figure supplement 1B–C*, left panel, lanes 3–10) or free yCAF1U or yCAF1X when in excess over DNA. Together, we conclude that DNA binding by yCAF1 is not necessary to mediate H3-H4 deposition in vitro. Addition of DNA consistently resulted in tetrasome deposition and release of H3-H4 from yCAF1. In no instance could we detect a yCAF1-H3-H4-DNA co-complex. Supporting this model, a yCAF1-DNA complex formed in the absence of H3-H4 migrated at a similar position on the native PAGE gel as the yCAF1-DNA complexes seen in the H3-H4 deposition reactions (*Figure 4—figure supplement 2C*).

To assess the contribution of the H3-H3′ tetramerization interface, we produced a yCAF1 complex bound to the H3M-H4 mutant and analyzed DNA deposition. This complex was deficient in histone deposition on DNA as seen by a lack of disome or tetrasome deposition. Instead, mostly non-specific binding is seen (*Figure 4C*, left panel, lanes 4–10). Apparently this defect arose due to a failure to release histones from yCAF1 (*Figure 4C*, right panel, lanes 3–9). Released yCAF1 would be expected to migrate as the free yCAF1 complex (*Figure 4C*, lane 12). While some free yCAF1 is seen at the highest concentration used (10 μM; *Figure 4C*, lane 10), the amounts of free yCAF1 were clearly lower than that seen in other DNA deposition assays. We also confirmed the deposition defect directly by extracting the yCAF1-DNA bands from the native PAGE gel followed by analysis

**Table 4.** DNA binding by yCAF1.

| Protein sample | DNA substrate | $K_D$ [μM] * | Hill coefficient |
|---|---|---|---|
| yCAF1 | 147 bp | 2.1 ± 0.1 | 2.0 ± 0.4 |
| | 84 bp | 2.3 ± 0.3 | 2.2 ± 0.3 |
| | 42 bp | 2.5 ± 0.5 | 2.2 ± 0.4 |
| | 17 bp | 5.1 ± 1.0 | 1.3 ± 0.2 |
| yCAF1U | 147 bp | 4.1 ± 1.9 | 4.3 ± 5.8 |
| yCAF1V | 147 bp | >10 | N.D. |
| yCAF1X | 147 bp | 5.7 ± 1. 7 | 2.9 ± 0.6 |
| yCAF1T | 147 bp | >10 | N.D. |

*Values determined from experiments using the 147, 84, 42 or 17 bp DNA fragment. Errors, where reported, correspond to the SEM value of three technical replicates.

**Table 5.** Sequence information on 10 bp DNA ladder (Promega). AT content (%) for all DNA fragments is 60%.

| Length (bp) | Sequence |
| --- | --- |
| 10 | GGACTATACT |
| 20 | GGACTATACTAGACATTGAC |
| 30 | GGACTATACTAGACATTGACGTGGTTGTAA |
| 40 | GGACTATACTAGACATTGACGTGGTTGTAAGATGATCATG |
| 50 | GGACTATACTAGACATTGACGTGGTTGTAAGATGATCATGTGTTAATGGC |
| 60 | GGACTATACTAGACATTGACGTGGTTGTAAGATGATCATGTGTTAATGGCAAGGTGAGTT |
| 70 | CATGATCATCTTACAACCACGTCAATGTCTAGTATAGTCCTACTCTGTGATATGGTTCTCTGTCGATGTA |
| 80 | GCCATTAACACATGATCATCTTACAACCACGTCAATGTCTAGTATAGTCCTACTCTGTGATATGGTTCTCTGTCGATGTA |
| 90 | AACTCACCTTGCCATTAACACATGATCATCTTACAACCACGTCAATGTCTAGTATAGTCCTACTCTGTGATATGGTTCTCTGTCGATGTA |
| 100 | ATGATCATCTAACTCACCTTGCCATTAACACATGATCATCTTACAACCACGTCAATGTCTAGTATAGTCCTACTCTGTGATATGGTTCTCTGTCGATGTA |

by SDS-PAGE. As expected, in the absence of DNA, the extracted yCAF1-H3-H4 band, which migrated close to top of the gel (* in *Figure 4A*, lane 11), showed yCAF1 and bound H3-H4 (*Figure 4—figure supplement 2D*, lane 1). With wild-type H3-H4, the DNA-bound complex (* in *Figure 4A*, lane 7), showed yCAF1 but no H3-H4 (*Figure 4—figure supplement 2D*, lane 2), showing that yCAF1 had released H3-H4. However with the H3M-H4 mutant, the DNA-bound yCAF1 complex (* in *Figure 4C*, lane 8) contained H3M-H4 (*Figure 4—figure supplement 2D*, lane 3), demonstrating that the deposition defect arose due to a failure to release H3-H4 from yCAF1. Taken together, these results provide compelling evidence that the ability of H3-H4 to tetramerize upon DNA deposition contributes to histone release from yCAF1.

## DNA-binding of yCAF1 is required for DNA synthesis-coupled nucleosome assembly

Our biochemical studies suggest a mechanism in which two yCAF1-H3-H4 complexes cooperatively bind to an extended DNA sequence element to assemble and deposit H3-H4 tetrasomes. We used a *Xenopus* egg high-speed egg extract (HSE) system to study the ability of yCAF1 mutant complexes to perform DNA synthesis-dependent chromatin assembly. To compare DNA-synthesis dependent and independent chromatin assembly, we used a plasmid with (pBS$_{uv}$) or without UV-damage (pBS$_o$), as described previously (*Ray-Gallet et al., 2007*). Indeed, *Xenopus* HSEs enabled us to follow chromatin assembly independently of the requirement for DNA synthesis when using pBS$_o$ (*Figure 5A*, upper panel, lanes 3, 6) and in a DNA synthesis-dependent manner when using UV-treated DNA pBS$_{uv}$ (*Figure 5A*, lower panel, lane 6).

To assess whether recombinant yCAF1 complexes are able to promote nucleosome assembly in a physiological context, we first immunodepleted the endogenous p150 from HSEs (*Figure 5—figure supplement 1*) and used these extracts for complementation assays. Such p150-depleted extracts were unable to assemble chromatin in a DNA synthesis-dependent fashion (*Figure 5B*, lane 4). Previous data show that these HSEs can be complemented with human p150 (*Quivy et al., 2001*). Here, we report that addition of increasing amounts of yCAF1 rescued DNA synthesis-dependent chromatin assembly (*Figure 5B*, lane 4–7). This experiment allowed us to establish limiting concentrations of yCAF1 that allow nucleosome assembly, as described previously (*Mello et al., 2002*). The different CAF1 mutants showed, to varying degrees, defects in DNA synthesis-coupled nucleosome assembly (*Figure 5—figure supplement 1B*) and specific conditions allowed us to discriminate for their efficiency (*Figure 5C,D*). The construct lacking the coiled-coil K/E/R domain (yCAF1T) showed residual nucleosome assembly activity as judged by the relative intensity of supercoiled product (*Figure 5C*). Constructs lacking the WHD (yCAF1U) or both the N-terminal region and WHD (yCAF1X), showed more deficient nucleosome assembly activity (*Figure 5C*). yCAF1V, a variant containing a deletion of both the coiled-coil and WHD of Cac1 showed the greatest assembly defect, as judged from the amount of supercoiled plasmid product (labeled I in *Figure 5C*). This defect of yCAF1V was even

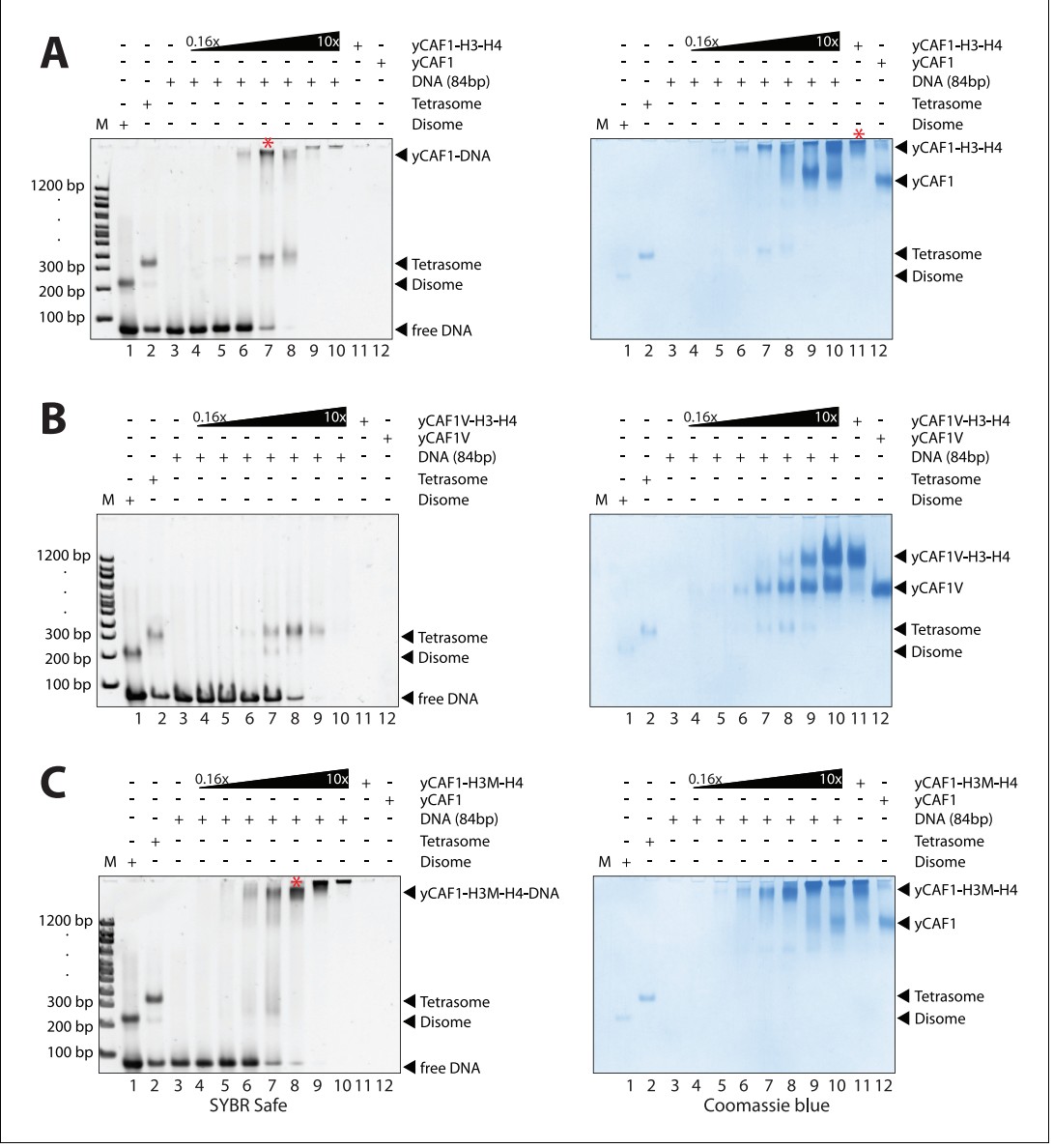

**Figure 4.** yCAF1 deposition of H3-H4. (**A**) EMSA showing tetrasome deposition on 84 bp DNA. Increasing amounts of yCAF1-H3-H4 (0.15, 0.3, 0.61, 1.25, 2.5, 5 or 10 μM) were mixed with 1 μM 84 bp DNA and the bands resolved by native PAGE. Gels were stained for DNA with SYBR Safe (left panel) and for protein with Coomassie (right panel). (**B**) As above but for yCAF1V-H3-H4. (**C**) As above but for yCAF1-H3M-H4 (H3M contains the L126R/I130R mutation). * indicates extracted gel bands that we analyzed by SDS-PAGE (*Figure 4—figure supplement 2D*). All EMSA experiments were repeated at least two times with consistency.

The following figure supplements are available for figure 4:

**Figure supplement 1.** EMSA analysis of H3-H4 deposition.
**Figure supplement 2.** EMSA analysis of H3-H4 deposition.

observable when increasing the yCAF1-DNA ratio such that all other mutants were able to almost fully compensate for the nucleosome assembly defect (*Figure 5D*). The mutants yCAF1T and yCAF1V contain a partially truncated PIP motif in which the first two residues of the conserved Q-x-x-I-x-x-F-F motif are absent (*Figure 1A*). As this motif is a PCNA binding site, we asked whether the

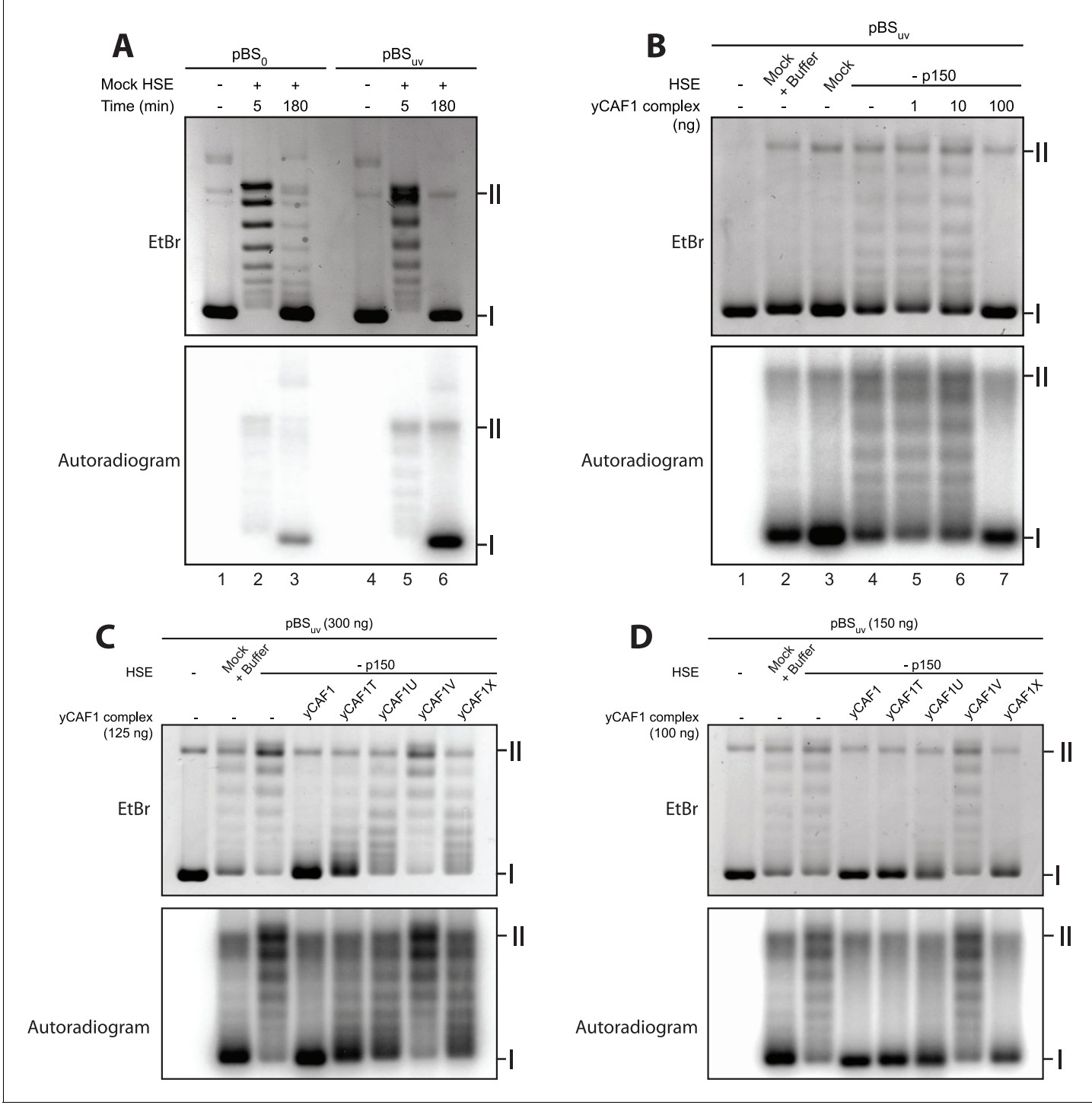

**Figure 5.** DNA-binding of yCAF1 is required for DNA synthesis-coupled nucleosome assembly. (**A**) Nucleosome assembly reactions with either non-UV-treated plasmid (pBS$_0$) or plasmid irradiated with UV (pBS$_{uv}$) in presence of [α−$^{32}$P]. After an incubation time of 5 or 180 min, DNA was extracted, resolved on an agarose gel and visualized by ethidium bromide staining (EtBr) or by autoradiography (Autradiogram). (**B**) 150 ng of pBSuv plasmid was incubated with p150-depleted HSE extracts and complemented with the indicated amounts of yCAF1. After 3 hr incubation, DNA was extracted, resolved on and agarose gel and visualized as above. (**C**) As in (**B**) but reactions were complemented with 125 ng of the different yCAF1 variants and the amount of pBSuv plasmid was increased to 300 ng (**D**) As in (**B**) but reactions were complemented with 100 ng of the different yCAF1 variants. (I) Supercoiled and (II) relaxed plasmid. All reactions were repeated two times with consistency.

The following figure supplement is available for figure 5:

*Figure 5 continued on next page*

*Figure 5 continued*

**Figure supplement 1.** Depletion of *Xenopus* p150 from HSE.

nucleosome assembly defect of these mutants is due interference with PCNA binding. A FLAG-pull-down experiment showed that yCAF1V was able to interact with PCNA, albeit to a lesser extent than yCAF1 wild-type (*Figure 5-supplemental figure 1C*). We therefore cannot fully exclude that the defective PIP motif contributes to the phenotype that we describe here. However it is clear that DNA binding is required for yCAF1 activity. The two DNA-binding domains were essential for CAF1 activity as the absence of either domain interfered to different degrees with DNA synthesis-dependent chromatin assembly.

## Interaction of yCAF1 with Asf1 and Mcm2

A current model suggests that upon DNA replication, parental H3-H4 are transiently maintained as dimers by Mcm2-Asf1 and reassembled into tetramers by CAF1 behind the replication fork (*Clément and Almouzi, 2015*; *Huang et al., 2015*; *Richet et al., 2015*). Asf1 is known to interact with the Cac2/p60 subunit of CAF1 possibly to enable histone transfer between the two chaperones (*Kim et al., 2016*; *Malay et al., 2008*; *Mello et al., 2002*). To assess a possible histone hand-over mechanism from Asf1-Mcm2 towards yCAF1, we systematically analyzed various combinations of these histone chaperones in the absence or presence of H3-H4 substrate by SEC-MALLS (*Figure 6*, *Table 6*). yAsf1 and yCAF1 did not interact under these conditions, presumably due to the transient nature of this interaction (*Malay et al., 2008*). In the presence of H3-H4, we observed yCAF1-H3-H4 and Asf1-H3-H4 but no higher-order complex (*Figure 6A*). Addition of a fivefold molar excess of preassembled Asf1-H3-H4 to yCAF1 resulted in appearance of a yCAF1-H3-H4 complex, due to transfer of H3-H4 from Asf1 to yCAF1 (*Figure 6A*). These data show that yCAF1 is able to receive H3-H4 from Asf1, but that there is no stable higher-order complex of these histone chaperones. An N-terminal fragment of Mcm2, spanning amino acid residues 1–160 and sufficient for H3-H4 binding, also did not interact directly with yCAF1 (*Figure 6B*, *Table 3*). In a direct competition assay, H3-H4 interacted preferentially with yCAF1 and Mcm2 migrated in its unbound form (*Figure 6B*). Incubation of a fivefold molar excess of Mcm2-H3-H4 with yCAF1 resulted in transfer of H3-H4 to yCAF1 (*Figure 6B*). Finally, addition of equimolar ratios of the three chaperones Mcm2 (1-160), yAsf1 and yCAF1 followed by addition of H3-H4 led to the formation of the yCAF1-H3-H4 and yAsf1-H3-H4-Mcm2 complexes. Of note, the molecular masses obtained are compatible with the model that a single H3-H4 dimer is bound to the complexes (*Table 6*).

## Discussion

In agreement with recent negative stain electron microscopy data (*Kim et al., 2016*), we found that yCAF1 forms an elongated heterotrimer containing a single copy of each subunit and that H3-H4 binds in a central position. Our deletion mapping is also in agreement with recent cross-linking MS experiments that indicate that the Cac1 subunit scaffolds interactions with Cac2, Cac3 and histones H3-H4 (*Kim et al., 2016*; *Liu et al., 2016*). As the isolated Cac1 subunit is able to interact with H3-H4 (*Liu et al., 2016*), our data are in agreement with the model that Cac1 contributes substantially to H3-H4 binding while Cac2 and Cac3 provide accessory interactions.

However, in contrast to previous findings (*Liu et al., 2012b*, *2016*; *Winkler et al., 2012*), we could not detect the presence of a H3-H4 tetramer bound to yCAF1 even when H3-H4 were supplied in excess and under conditions that favor tetramerization (*Figure 1—figure supplement 2C*). Previous data are based on fluorescence titration or FRET experiments which show that yCAF1 binds two copies of H3-H4 or a covalently crosslinked H3-H4 tetramer with high ($K_D^{app}$ ~5 nM) binding affinity (*Liu et al., 2012b*, *2016*; *Winkler et al., 2012*). Our data suggest that high-affinity histone binding by yCAF1 is driven, at least in part, by the highly polar DNA-binding surface of H3-H4. Considering that the DNA-binding surface is maintained in the crosslinked H3-H4 tetramer, it is not too surprising that this substrate binds with high affinity if it can be accommodated sterically.

We found that yCAF1 can receive H3-H4 from Asf1 or Mcm2 without forming a stable higher-order complex among these histone chaperones (*Figure 6*). Thus, we propose that the mechanism

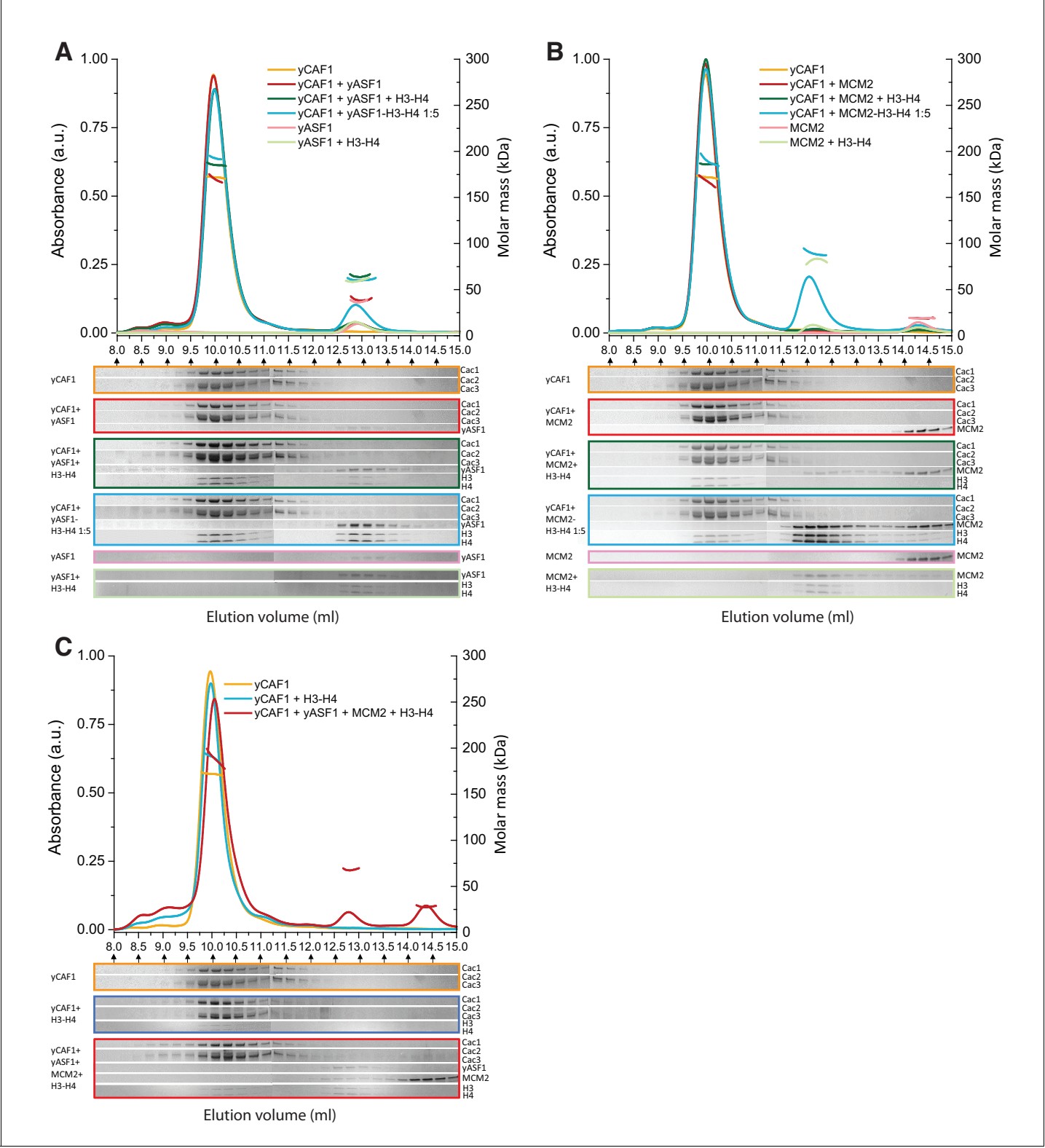

**Figure 6.** Competition of yCAF1 with yAsf1 or Mcm2 for H3-H4 binding. SEC-MALLS analysis of complexes formed upon mixing of up to three histone chaperones with H3-H4. In all experiments, lines correspond to the UV$_{280nm}$ traces of the eluting complex. Dots correspond to the molar mass measurements. Eluting fractions were analyzed by SDS-PAGE and the relevant areas of the gels are displayed below each chromatogram using the corresponding color code. When there are two or more proteins in the mixture, the '+' in the labeling indicates the order in which the samples were mixed together (eg. yCAF1 + yAsf1 +H3-H4 indicates that yAsf1 was added to yCAF1 followed by addition of H3-H4). The final protein concentration

*Figure 6 continued on next page*

*Figure 6 continued*

used was 20 μM for all proteins except where preformed yAsf1-H3-H4 and Mcm2-H3-H4 were supplied in 5-fold molar excess to yCAF1 (blue lines in A and B). (**A**) H3-H4 competition experiments of yCAF1 with yAsf1. (**B**) H3-H4 competition experiments of yCAF1 with Mcm2. (**C**) H3-H4 competition experiments of yCAF1 in the presence of yAsf1 and Mcm2.

of H3-H4 tetrasome assembly by yCAF1 requires yAsf1- or Mcm2-dependent transfer of H3-H4 towards yCAF1 (*Figure 7*). Although the Cac1 subunit contains a reported DNA-binding WHD that binds to 10–16 bp DNA (*Zhang et al., 2016*), this domain is not sufficient for high-affinity DNA binding. In agreement, point mutations that abolish DNA binding of the isolated WHD do not abolish DNA binding by yCAF1 (*Zhang et al., 2016*). Nevertheless, the WHD clearly contributes to overall yCAF1 activity and might play a role in transcriptional silencing activity and the DNA damage response presumably by aiding recruitment of yCAF1 to replication forks (*Liu et al., 2016*; *Zhang et al., 2016*).

High affinity DNA binding by yCAF1 requires a second DNA-binding element that we locate to the amino acid residues 135–230 of Cac1, a region containing the K/E/R domain and is predicted to form a coiled-coil. This region is also critical in human p150, as a deletion mutant spanning the equivalent region (amino acids 311–445) abolishes chromatin assembly activity (*Kaufman et al., 1995*). We identified two mutants, yCAF1T and yCAF1V, that were deficient in DNA interaction but that retained H3-H4-binding activity. Counterintuitively, the WHD appears to be inert for DNA binding in yCAF1T (*Figure 3A*). Previous hydrogen/deuterium exchange (HX) data indicate that H3-H4 binding results in structural rearrangements in Cac1 as evidenced by increased HX for amino acids 550–591 (*Liu et al., 2016*). Thus, H3-H4 binding to yCAF1 could result in unmasking of the DNA-binding activity of the WHD. However DNA synthesis-dependent chromatin assembly requires the presence of both the coiled-coil K/E/R and WHD DNA-binding domains.

yCAF1 binds preferentially to DNA of 40–80 bp length and we find that binding to such extended DNA elements is cooperative. Assembly of H3-H4 tetrasomes requires a DNA substrate of similar

**Table 6.** Summary of SEC-MALLS data. Column labeling: $V_e$ (elution Volume); $MM_{SLS}$ (Molar masses determined by SEC-MALLS); $MM_{th}$ (theoretical molar mass calculated). When there are more than two proteins in the injected sample, '+' indicates the mixing order. In sample 4, a five-fold molar excess of a preformed yAsf1-H3-H4 complex was incubated with yCAF1 before injection. In sample 9, a five-fold molar excess of a preformed MCM2-H3-H4 complex was incubated with yCAF1. The errors reported are the residual standard deviations of the observed data from the fitted values calculated using Astra.

| | Peak 1 | | | Peak 2 | | | Peak 3 | | |
|---|---|---|---|---|---|---|---|---|---|
| Sample | $V_e$ (ml) | $MM_{sls}$ (kDa) | $MM_{th}$ (kDa) | $V_e$ (ml) | $MM_{sls}$ (kDa) | $MM_{th}$ (kDa) | $V_e$ (ml) | $MM_{sls}$ (kDa) | $MM_{th}$ (kDa) |
| yCAF1 | 9.96 | 172.1 ± 0.1 | 174.0 | – | – | – | – | – | – |
| yCAF1 + yAsf1 | 9.96 | 171.1 ± 0.6 | 174.0 | 12.91 | 38.6 ± 0.2 | 31.6 | – | – | – |
| yCAF1 + yAsf1 + H3 H4 | 10.00 | 185.3 ± 0.1 | 200.7 | 12.85 | 64.3 ± 0.2 | 58.3 | – | – | – |
| yCAF1 + yAsf1-H3-H4 1 :5 | 9.98 | 192.9 ± 0.2 | 200.7 | 12.88 | 60.1 ± 0.1 | 58.3 | – | – | – |
| yAsf1 | – | – | – | 12.96 | 36.3 ± 0.1 | 31.6 | – | – | – |
| yAsf1 + H3 H4 | – | – | – | 12.86 | 58.7 ± 0.2 | 58.3 | – | – | – |
| yCAF1 + MCM2 | 9.95 | 174.1 ± 0.7 | 174.0 | – | – | – | 14.32 | 19.1 ± 0.5 | 17.6 |
| yCAF1 + MCM2 + H3 H4 | 9.97 | 185.9 ± 0.1 | 200.7 | – | – | – | – | – | – |
| yCAF1 + MCM2-H3-H4 1 :5 | 9.98 | 190.5 ± 0.7 | 200.7 | 12.09 | 90.8 ± 0.4 | 71.0 | – | – | – |
| MCM2 | – | – | – | – | – | – | 14.32 | 18.8 ± 0.1 | 17.6 |
| MCM2 + H3 H4 | – | – | – | 12.16 | 82.6 ± 0.4 | 71.0 | – | - | – |
| yCAF1 + H3 H4 | 9.97 | 191.3 ± 0.4 | 200.7 | – | – | – | – | – | – |
| yCAF1 + yAsf1+Mmc2 + H3 H4 | 10.05 | 188.8 ± 0.9 | 200.7 | 12.79 | 67.5 ± 0.1 | 75.9 | 14.38 | 27.4 ± 0.1 | 17.6 |

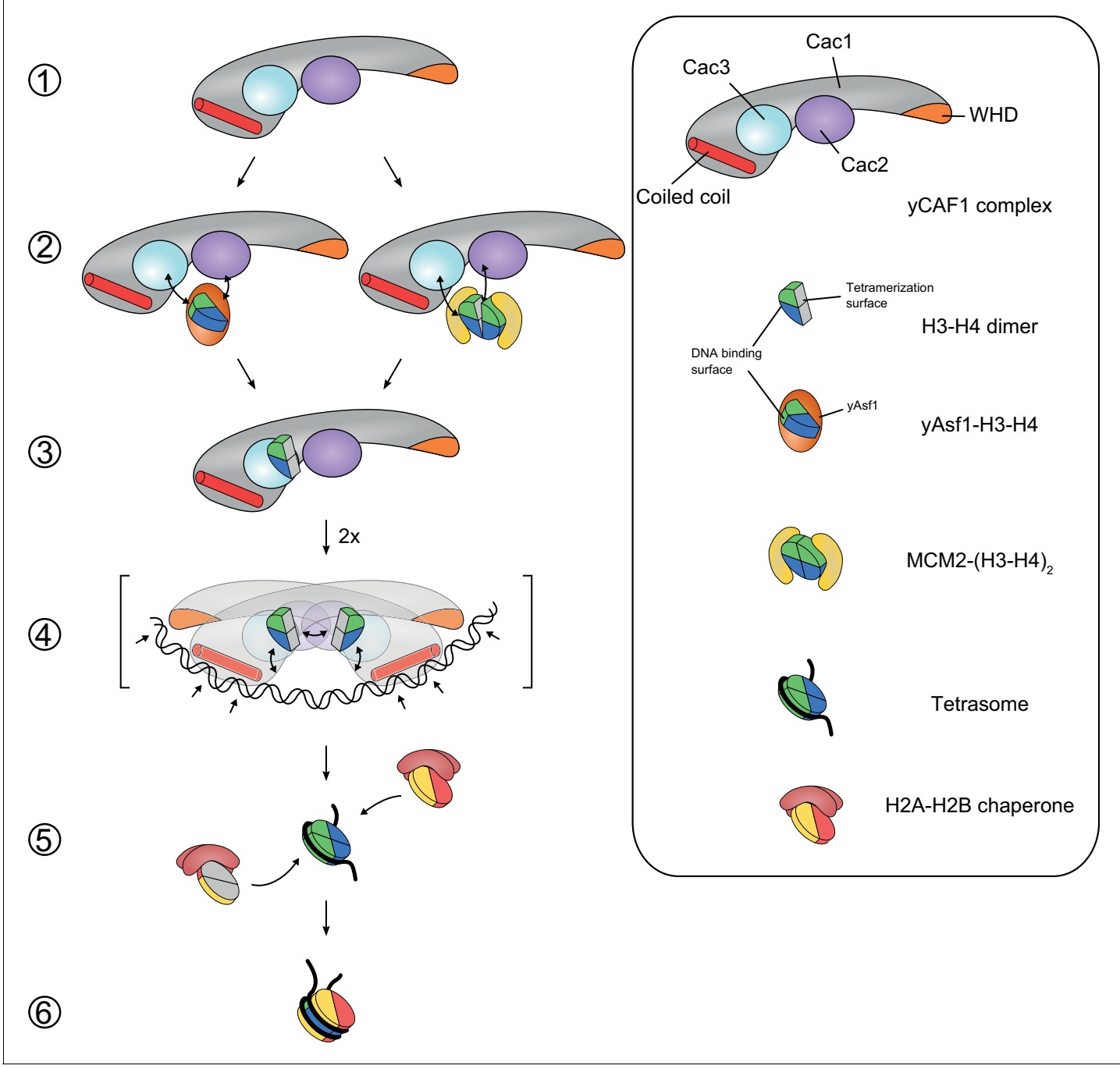

**Figure 7.** Model for yCAF1 recruitment and H3-H4 deposition. Free monomeric yCAF1 (step 1) is loaded with dimeric H3-H4 through association of yAsf1 with the Cac2 subunit. Alternatively, loading can occur through hand over of H3-H4 from Mcm2 (step 2). yCAF1 binds the histones via their DNA binding and oligomerization surfaces (step 3). During DNA synthesis, two yCAF1-H3-H4 complexes bind cooperatively to an extended DNA element >50 bp (step 4) to deposit H3-H4 dimers and form tetrasomes. The WHD (orange) and coiled-coil (red) DNA-binding domains of yCAF1 are required for deposition of H3-H4 tetramers. The requirement of an extended free DNA region together with PCNA interaction may direct yCAF1 activity to replication forks. H2A-H2B chaperones like NAP1 or FACT recognize the tetrasome intermediate and deposit two copies of H2A-H2B (step 5) to form a complete nucleosome (step 6).

length ~60–80 bp (*Luger et al., 1997*). Cooperative binding usually requires some form of interaction between binding partners but we do not find evidence for yCAF1 or yCAF1-H3-H4 dimerization in solution. Previous studies have indicated, that the isolated Cac1 subunit and the p150 ortholog, have a tendency to dimerize in the absence of their binding partners (*Gérard et al., 2006*; *Liu et al., 2016*; *Quivy et al., 2001*; *Winkler et al., 2012*). A C-terminal fragment of Cac1, spanning amino acid residues 385–606, is sufficient to bind H3-H4 and to promote deposition of a H3-H4 tetramer onto DNA (*Liu et al., 2016*), but the physiological relevance of dimerization of isolated Cac1/p150 remains unclear. yCAF1 binds cooperatively with DNA and when loaded with H3-H4 heterodimers, this results in tetrasome deposition. Our model is that histone tetramerization gives directionality to the deposition reaction as it is required for histone release from yCAF1. H3-H4 tetramers interact with higher binding affinity (~1 nM) with DNA as compared to yCAF1 (~5 nM) (*Andrews et al., 2010*; *Winkler et al., 2012*). A H3M-H4 mutant is expected to show decreased DNA-binding affinity as tetramerization stabilizes H3-H4 on DNA. However the H3M-H4 mutant retains high affinity (~5.5 nM) yCAF1 binding (*Winkler et al., 2012*), likely explaining the failure to release from yCAF1 in a DNA deposition assay (*Figure 4C* and *Figure 4—figure supplement 2D*).

yCAF1 has a preference for regular B-DNA as we failed to detect binding to nucleosome core particles that are known to contain DNA with unusual conformational parameters (*Luger et al., 1997*). An extended nucleosome-free DNA region of ~250 bp is present behind and ~300 bp ahead of the replication fork (*Gasser et al., 1996*; *Sogo et al., 1986*). Nucleosome assembly occurs as soon as DNA of sufficient length to wrap around the histone octamer core has passed through the replication machinery (*Sogo et al., 1986*). Thus it is conceivable that the DNA-binding preferences of yCAF1 described here allow yCAF1 to 'sense' regions of extended nucleosome-free DNA and promote nucleosome assembly preferentially onto the nucleosome-free region of replicating DNA. In contrast, assembled chromatin, which contains nucleosomes separated by small linker DNA (~15–20 bp on average in yeast), would be expected not to allow the yCAF1 binding thereby preventing its action (*Radman-Livaja and Rando, 2010*). This could explain why yCAF1 does not promote significant nucleosome assembly in the absence of ongoing DNA replication. Not unlike what we propose here for yCAF1, direct DNA binding by HIRA and Rtt106, histone chaperone complexes involved in deposition of different H3 variants, has been proposed (*Liu et al., 2010*; *Ray-Gallet et al., 2011*; *Schneiderman et al., 2012*).

PCNA has long been reported to act as the key recruitment vehicle for CAF1 at replication forks (*Moggs et al., 2000*; *Shibahara and Stillman, 1999*; *Zhang et al., 2000*). Yeast Cac1 contains one PCNA binding site (PIP) that is conserved from yeast to human p150 and is required for interaction with PCNA (*Krawitz et al., 2002*). We suggest that yCAF1 recruitment to sites of DNA replication requires cooperation between DNA binding and PCNA interaction. Such dependencies are frequently seen in chromatin regulators that utilize high-affinity nucleic acid-binding domains coupled to low-affinity binding modules that recognize histone modifications (*Ptashne, 2009*). The high-affinity interactions thereby reduce the entropic penalty of bringing the weakly binding modules and their substrate together and expand the dynamic range for local recruitment by increasing affinity up to several-fold (*Ptashne, 2009*).

Our data are consistent with a model where two yCAF1-H3-H4 complexes cooperatively bind to an extended DNA sequence element to deposit H3-H4 tetrasomes (*Figure 7*). In solution, H3-H4 tetramerization in the yCAF1 complex is not permitted, possibly because residues in the H3-H3' dimerization interface are occluded. Furthermore, considering that H3-H4 binding to yCAF1 or DNA is mutually exclusive, we propose that yCAF1 interacts with the DNA-binding surface of H3-H4. This model is also consistent with our competition experiments with Mcm2 (*Figure 6*). All the yCAF1 variants analyzed allowed tetrasome assembly using purified components, but the presence of the DNA-binding domains is required for DNA synthesis-coupled nucleosome assembly (*Figure 5*). H3-H4 tetramerization stabilizes the H3-H4 dimers on DNA and triggers the release from yCAF1 (*Figure 4C*). This model is in agreement with data showing that H3-H4 dimers do not assemble as tetramers prior to deposition onto DNA (*Benson et al., 2006*; *Tagami et al., 2004*) and puts forward the importance of a series of coordinated transitions that require the unique DNA-binding properties of yCAF1. Together, we propose a model for DNA synthesis-coupled chromatin assembly in which (i) yCAF1 binds to a single H3-H4 dimer, (ii) two yCAF1-H3-H4 complexes can bind to nascent DNA in a concerted manner enabling H3-H4 tetramer formation on DNA, which (iii) results in release of histones from yCAF1. The DNA substrate requirements of the reaction could fit nicely

with the length of Okazaki fragments during replication on the lagging strand, the periodic sizing of which also depends on yCAF1-dependent histone deposition (*Smith and Whitehouse, 2012*), and would also be relevant during nucleotide excision repair. Future work, in vivo and in other organisms, should address how the mechanistic insights provided here are linked to inheritance of chromatin-based information.

## Materials and methods

### Construct design, generation of baculoviruses and expression in Hi5 cells

The three yCAF1 subunits Cac1, Cac2 and Cac3 from *Saccharomyces cerevisiae* were amplified by PCR from pPK133, pPK160 and pPK134 and cloned into the pIDC, pIDK and pFL vectors for the Multibac system using InFusion cloning (*Trowitzsch et al., 2010*). Cac3 contained a N-terminal decahistidine tag, Cac2 was FLAG-tagged, both containing TEV protease recognition sites for removal of the tags. N-terminal Cac1 deletions were generated by PCR of the pIDC-Cac1 plasmid and by using forward primers carrying a NdeI site, amplifying the shortened version of Cac1. The corresponding, NdeI containing reverse primer amplified the plasmid backbone. PCR products were cut with NdeI, purified and re-ligated.
Forward primers:
pIDC_Cac1_NdeI_129_fwd 5' ATATCATATGAAGAGAGAACTTTCCTCATCG 3'
pIDC_Cac1_NdeI_230_fwd 5' ATATCATATGATTGGTAACTTCTTTAAAAAACTAAGCG 3'
Reverse primer:
pIDC_BB_NdeI_START_rev 5' ATATCATATGCGGACCGGGATCCGC 3'
The WHD deletion was generated using a similar approach by using the following primers carrying XhoI sites:
pIDC_Cac1_494_XhoI_rev 5' ATATCTCGAGTTACGATGTTTTGGGTTC 3'
pIDC_BB_XhoI_fwd 5' ATATCTCGAGGGCCTACGTCGACGAG 3'

Cac1 truncations were designed to remove regions containing predicted disorder and considering limited proteolysis experiments (*Figure 1—figure supplement 1*). Baculovirus generation and expression in Hi5 cells was carried out as described before (*Trowitzsch et al., 2010*).

### yCAF1 expression and purification

1–2 l Hi5 cell culture expressing yCAF1 constructs were harvested by centrifugation (800 g, 20 min, 4°C) and resuspended in 100 ml lysis buffer (20 mM TRIS pH 7.8, 500 mM NaCl, 5 mM Imidazole, 0.1% NP-40, 0.5 mM TRIS (2-carboxyethyl)phosphine hydrochloride (TCEP), containing protease inhibitor tablets (Roche, Switzerland) and Benzonase. Cells were lysed by sonication for 30 s on ice followed by centrifugation at 30000 g for 45 min at 4°C. The soluble lysate was loaded onto a 5 ml HisTrap FF column (GE Healthcare, UK), pre-charged with $Co^{2+}$ ions and equilibrated in lysis buffer using a peristaltic pump at 4°C. After loading, the column was washed in with 10 column volumes (CV) of lysis buffer, followed by 40 CV of wash buffer (20 mM TRIS pH 7.8, 500 mM NaCl, 5 mM Imidazole, 0.5 mM TCEP). Bound protein was eluted with buffer containing 20 mM TRIS pH 7.8, 300 mM NaCl, 500 mM Imidazole 0.5 mM TCEP and subsequently diluted with 20 mM TRIS pH 7.8, 0.5 mM TCEP to give a final NaCl concentration of 150 mM NaCl. The sample was loaded onto a 5 ml HiTrap Q column (GE Healthcare) and subsequently connected to an AKTA Purifier FPLC system for washing in Q-150 buffer (20 mM TRIS pH 7.8, 150 mM NaCl, 0.5 mM TCEP) and elution using a 20 CV gradient of buffer Q-1000 (20 mM TRIS pH 7.8, 1 M NaCl, 0.5 mM TCEP). yCAF1-containing fractions were identified by SDS-PAGE, pooled and concentrated before injection onto a Superdex 200 16/60 size exclusion column (GE Healthcare) equilibrated in SEC buffer (20 mM BisTRIS pH 6.5, 500 mM NaCl, 0.5 mM TCEP). Protein purity was assessed and the protein-containing fractions were pooled and concentrated to 8–23 mg ml$^{-1}$ using Amicon centrifuge filter units (100 kDa cutoff). Concentrated protein was maintained at 4°C or flash frozen in liquid nitrogen and stored at −80°C.

### Expression, purification and reconstitution of Histones H3 and H4

Recombinant *Xenopus laevis* histones H3 and H4 were expressed, purified and refolded according to standard procedures (*Luger et al., 1999*). The H3 tetramerization mutant (H3M), containing the

point mutations L126R and I130R which disrupt tetramer formation (*Winkler et al., 2012*), was created by directed mutagenesis from the wild type plasmid using primers
5' GCTGGCCCGCAGAAGGCGAGGCGAGAGG 3' and
5' CCTCTCGCCTCGCCTTCTGCGGGCCAGC 3' for I130R and
5' CAAGGACATCCAGCGGGCCCGCAGAATCC 3' and
    5' GGATTCTGCGGGCCCGCTGGATGTCCTTG 3' for L126R and verified by DNA sequencing. H3M was expressed, purified from inclusion bodies following the same procedure as used for the wild type and the presence of the mutation verified by LC-MS.

H3-H4 were assembled by dissolving equimolar amounts of each lyophylized histone in unfolding buffer (20 mM TRIS pH 7.5, 7 M Guanidinium chloride, 5 mM $\beta$-mercaptoethanol). After mixing H3-H4, the sample was incubated for one hour, followed by dialysis for 16–18 hr at 4°C against refolding buffer (10 mM TRIS, pH 7.5, 2 M NaCl, 1 mM Na-EDTA, 5 mM $\beta$-mercaptoethanol). The sample was then run in refolding buffer on an equilibrated HiLoad 16/60 Superdex 75 column (GE Healthcare). Aliquots were stored in 50% glycerol at −20°C.

## yCAF1-H3-H4 complex formation

To prepare yCAF1-H3-H4 complexes, reconstituted H3-H4 tetramers were added at twofold molar excess to yCAF1. The samples were incubated for up to 120 min on ice before loading on a Superdex 200 10/300 GL column (GE Healthcare) in SEC buffer. Protein-containing fractions were pooled and concentrated to 20–30 mg ml$^{-1}$ using Amicon centrifuge filter units (100 kDa cutoff). Concentrated protein was maintained at 4°C or flash frozen in liquid nitrogen and stored at −80°C.

## Salt deposition of histones onto DNA and EMSA

For salt-deposition of histones, we used standard protocols (*Dyer et al., 2004*). Briefly, H3-H4 tetramers or H3M-H4 dimers were mixed with 84 bp or 147 bp DNA derived from the canonical Widom sequence in 20 mM TRIS, pH 8.0, 2 M NaCl, 1 mM EDTA, 1 mM DTT and dialyzed against 1.5 M NaCl buffer for 2–3 hr at 4°C. The samples were then transferred into consecutively lower (first 1 M, 0.5 M and then 0.25 M) NaCl concentration buffer for 2 hr each with the second-last dialysis being an overnight step. Samples were then incubated at 37°C for 15 min and then maintained on ice prior to analysis. For analysis of DNA-binding by the EMSA, yCAF1 or yCAF1-H3-H4 were incubated with the 84 bp or 147 bp DNA in EMSA buffer (500 mM NaCl, 20 mM BisTRIS, pH 7.8, 0.5 mM TCEP, 5% glycerol). The samples were maintained on ice for 30 min and then heat shifted at 37°C for 5 min prior to analysis. The binding reactions were analyzed on a 6% native 1x TRIS-Glycine (250 mM TRIS, 1.92 M glycine, pH 8.3) Mini-PROTEAN (Bio-rad, Hercules, CA) polyacrylamide gel using 1x TRIS-Glycine running buffer. The gel was stained with SYBR Safe (Thermo Fisher Scientific, Waltham, MA) to visualize DNA-bound complexes or Coomassie Blue for protein staining. Band intensities were quantified by ImageJ (Version 1.51) and the data analyzed by using the Origin software (Version 9.3) using a Hill equation binding model.

## Sample extraction from native PAGE gels

For extraction of protein/DNA bands from native PAGE gels, the bands indicated with an * in *Figure 4* were cut out and mechanically homogenized using a syringe with a 1.2 mm Ø needle. After addition of 1x SDS loading buffer the samples were boiled, spun down and 40 μl loaded on SDS-PAGE for analysis.

## Multi angle laser light scattering and SAXS analysis

Size-exclusion chromatography was performed at a flow rate of 0.5 ml min$^{-1}$ on a Superdex 200 Increase 10/300 GL column equilibrated in SEC-MALLS buffer (20 mM Bis-TRIS, pH 6.5, 500 mM NaCl, 1 mM DTT or TCEP) at 21°C. A 30 μl sample of yCAF1 ± H3-H4 at 2–10 mg ml$^{-1}$, previously incubated on ice for 10 min to 3 hr, was injected onto the column and multi angle laser light scattering was recorded with a laser emitting at 690 nm using a DAWN-EOS detector (Wyatt Technology-Corp. Santa Barbara, CA). The refractive index was measured using a RI2000 detector (Schambeck SFD, Germany). The molecular weight was calculated form differential refractive index measurements across the center of the elution peaks using the Debye model for protein using ASTRA

software version 6.0.5.3. To verify the stoichiometry of the yCAF1-H3-H4 complex, 1:1 and 1:2 ratios of yCAF1 to H3-H4 dimers were tested.

X-ray scattering data were collected using an inline HPLC setup, at the Bio-SAXS beamline (BM29) of the European Synchrotron Radiation Facility (*Pernot et al., 2013*). Inline size-exclusion chromatography was performed at a temperature of 10°C using a Superdex 200 Increase 10/300 GL column equilibrated in SEC-MALLS buffer. Data were collected with a photon-counting Pilatus 1M detector at a sample-detector distance of 2.86 m, a wavelength of $\lambda$ = 0.991 Å and an exposure time of 1 s/frame. A momentum transfer range of 0.008 to 0.47 $\text{Å}^{-1}$ was covered (q = $4\pi \sin\theta/\lambda$, where $\theta$ is the scattering angle and $\lambda$ the X-ray wavelength). Data collected across the peak were subtracted from buffer scattering and the frames showing a constant radius of gyration (*Rg*) were merged for further analysis. *Rg* values were obtained from the Guinier approximation *sRg* <1.3 using Primus (*Petoukhov et al., 2012*). Distance distribution functions *p(r)* and the Porod volumes *Vp* were computed from the entire scattering curve using GNOM (*Petoukhov et al., 2012*). The program DAMMIN (*Svergun, 1998*), was used to generate 40 low-resolution *ab initio* shape reconstructions. To select the most typical *ab initio* model of the complex and estimate its possible conformational space, these reconstructions were pairwise aligned and averaged using DAMAVER (*Volkov and Svergun, 2003*). The model with the lowest mean value of normalized spatial discrepancy (NSD) was selected as the most typical reconstruction. To assess the resolution and reliability of the reconstructions we used the Fourier Shell Correlation (FSC) approach as implemented in SASRES (*Tuukkanen et al., 2016*).

## Native mass spectrometry

Prior to non–denaturing MS analyses, 50 µl of yCAF1 or yCAF1-H3-H4 were buffer exchanged into 250 mM ammonium acetate, pH 6.8 using a Superdex 200 3.2/300 column mounted on an ÄKTAmicro system (GE Healthcare Life Sciences). The buffer of H3-H4 tetramers was exchanged into 1 M ammonium acetate pH 7.0. The exchange did not affect the complex integrity, as judged from the SEC elution profile. For all the measurements, 2–4 µl of sample were loaded into nanoflow platinum-coated borosilicate electrospray capillaries (Thermo Electron SAS, France). Protein ions were generated using a nanoflow electrospray (nano-ESI) source and MS analyses carried out on a quadrupole time-of-flight mass spectrometer (Q-TOF Ultima, Waters Corporation, U.K.). The instrument was modified for the detection of high masses (*Sobott et al., 2002*; *van den Heuvel et al., 2006*). The following instrumental parameters were used: capillary voltage = 1.2–1.3 kV, cone potential = 40 V, RF lens-1 potential = 40 V, RF lens-2 potential = 1 V, aperture-1 potential = 0 V, collision energy = 30–140 V, and microchannel plate (MCP) = 1900 V, ToF pressure $\approx 8 \times 10^{-6}$ mbar. For collision induced dissociation experiments, the collision voltage was increased up to 210 V, and collision cell pressure was $\approx 2 \times 10^{-4}$ mbar. All mass spectra were calibrated externally using a solution of cesium iodide (6 mg/mL in 50% isopropanol) and were processed with the Masslynx 4.0 software (Waters Corporation) and with Massign software package (*Morgner and Robinson, 2012*) with minimal smoothing and no background subtraction.

To calculate the molecular mass (M) and to estimate the standard deviation of the measurement, we followed a procedure described previously (*McKay et al., 2006*). Briefly, two neighboring *m/z* values ($M/z_1$ and $M/z_2$) are determined experimentally (x and y) and two equations are written ($M/z_1 = x$ and $M/z_2 = y$). Since $z_1 = z_2$–1, the equations are solved to determine M, $z_1$ and $z_2$ using the MassLynx software (Waters). The program takes several combinations of neighboring *m/z* values to determine distinct M values of a macromolecule. Using these values, a mean value of M and its standard deviation are calculated. The M values were determined from *m/z* values corresponding to the left edge of the peaks. These values provide the 'least-adducted' M of the noncovalent complexes (*McKay et al., 2006*) and are reported in *Table 1* and *Table 3*.

## Analytical Ultracentrifugation

Analytical centrifugation was performed using an Optima XL-A analytical ultracentrifuge (Beckman Coulter, Brea, CA) with an AN-60 Ti rotor. For sedimentation equilibrium experiments, six-channel cells were used, and data were acquired at a resolution of 0.001 cm with twenty replicates at a temperature of 4°C. Reference cells were loaded with buffer 20 mM Bis-TRIS (pH 6.5), 0.2 M NaCl, and 0.5 mM TCEP. Absorbance at 280 nm was used to monitor concentration gradients. yCAF1

concentrations were 0.5, 1 and 3.2 µM, and the speeds were 7.000, 10.000, and 40.000 rpm. yCAF1-H3-H4 concentrations were 0.5, 1 and 3.2 µM, and the concentration distribution was measured at identical rotor speeds as yCAF1. Samples were determined to have reached equilibrium when scans taken 4 hr apart showed no systematic differences. The data were analyzed with the program *Win-Nonlin*. For sedimentation velocity experiments the purified yCAF1 complexes were loaded into two- sector centerpieces, and buffer 20 mM Bis-TRIS (pH 6.5), 0.2 M NaCl, and 0.5 mM TCEP was used for the reference chamber. Experiments were performed at 42,000 rpm and 4°C. Data were collected at a wavelength of 280 nm, using a spacing of 0.003 cm, with one average in the continuous scan mode. No time delay was used, allowing traces to be collected every ~1 min. Sedimentation coefficients were corrected to standard conditions (20°C, in water) using DCDT+ (version 2.4.3) (*Philo, 2006*; *Stafford, 1992*). The sedimentation velocity data were analyzed to obtain the $g$ ($s^*$) distribution of the sample using DCDT+. The $g(s)$ distributions were further analyzed to obtain the apparent molecular weight of the sample using DCDT+. The protein partial specific volumes were calculated from the amino acid composition to 0.722 ml g$^{-1}$ (yCAF1) and 0.733 ml g$^{-1}$ (yCAF1-H3-H4) and solvent density was calculated through summation of the contribution of buffer components to 1.009 g cm$^{-3}$ at 4°C using the program SEDNTERP. Molecular mass was determined using Beckman software provided as an add-on to Origin version 3.8. All nine data sets were analyzed using a global fit procedure based on a model describing an ideal non-interacting single component system, with local parameters for reference concentrations and base-line offsets and global parameters for the molecular weight. Best-fits were determined through visual inspection of the residuals (*Figure 2—figure supplement 1*). To compare the consistency of the hydrodynamic parameters determined from SAXS and AUC, we determined a theoretical sedimation coefficient ($S_{th}$) from the SAXS beads model by using the program WinHydroPro++ (*Ortega et al., 2011*). Input parameters including solvent density, solvent viscosity and partial specific volume were determined using SEDNTERP as described above. The temperature was 4°C and the theoretical molecular mass was calculated from the primary sequence. The radii of atoms were set to the same values as that obtained from the SAXS bead models.

## Protein sample preparation for Asf1 and Mcm2 competition experiments

Recombinant *Drosophila melanogaster* histones H3-H4 (identical to human histones H3.2) and *Homo sapiens* Mcm2(1–160) were purified as described previously (*Richet et al., 2015*). Recombinant full length *Saccharomyces cerevisiae* Asf1 was produced and purified using the same protocol as for Mcm2 except that the HisTrap column was replaced by a nickel-nitrilotriacetic acid (Ni-NTA) column (Qiagen, Germany). The flow through was then loaded on an anion exchange column Resource Q (GE Healthcare) and Asf1 eluted using a buffer with 50 mM TRIS-HCl and 1 M NaCl. The elution buffer was replaced by 50 mM TRIS-HCl pH 7.5 storage buffer using an Amicon device (Millipore, Billerica, MA) and an YM10 regenerated cellulose membrane (Millipore). For molar mass determination, purified proteins were analyzed using SEC-MALLS as described previously (*Richet et al., 2015*). yAsf1 and yCAF1 were mixed in equimolar ratios (final concentration of both chaperones 20 µM) in 10 mM TRIS pH 7.5, 0.5 M NaCl, 0.5 mM TCEP (final volume 110 µL). To prevent H3-H4 aggregation a specific order of addition was maintained during sample setup (histones added last). Samples were incubated at 4°C overnight prior to injection of 100 µl of into a Superdex 200 Increase 10/300 GL column (GE Healthcare) equilibrated in 10 mM TRIS pH 7.5, 0.5 M NaCl, 0.5 mM TCEP at a flow rate of 0.5 ml.min$^{-1}$. Multi angle laser light scattering was recorded with a laser emitting at 690 nm using a DAWN-TREOS detector (Wyatt TechnologyCorp. Santa Barbara, CA). The refractive index was measured using a T-rEX detector (Wyatt technology. Santa Barbara, CA). The molecular weight was calculated from differential refractive index measurements across the center of the elution peaks using the Debye model for protein using ASTRA software version 6.1.7.13.

## DNA synthesis-dependent nucleosome assembly assays

*Xenopus* High-Speed Egg extract (HSE) preparation and chromatin assembly assays were prepared as described previously (*Ray-Gallet and Almouzni, 2004*). After removal of the jelly coat by cysteine treatment, *Xenopus laevis* eggs were rinsed in extraction buffer (10 mM KOH-HEPES pH 7.8, 70 mM KCl, 5% sucrose, 0.5 mM dithiothreitol (DTT) and protease inhibitors) and centrifuged at 150,000 g

for 1 hr at 4°C. The clear ooplasmic fraction was collected, aliquoted and stored at −80°C. Depletions were done by the addition of p150 antibody (*Quivy et al., 2001*) coupled to protein A-Sepharose slurry (CL-4B; Amersham Biosciences, UK) to HSE for 1 hr at 4°C on a rotating wheel. A pBS plasmid (Stratagene, La Jolla, CA) was used to perform the chromatin assembly reaction, damaged by UV-C (500 J/m$^2$) (named pBS$_{UV}$) or not (named pBS$_0$). 10 μL of HSE (depleted or mock depleted) was added to 150 ng or 300 ng of pBS$_{UV}$ or pBS$_0$ in a buffer containing 5 mM MgCl$_2$, 40 mM KOH-HEPES pH 7.8, 0.5 mM DTT, 4 mM ATP, 40 mM phosphocreatine, 2.5 μg of creatine phosphokinase and 5 μCi of [α-$^{32}$P]dCTP in a final volume of 25 μl. The reaction was incubated at 23°C for 3 hr. After 5 min, yCAF1 complexes or buffer were added to the reaction. Chromatin assembly was stopped by the addition of 25 μl of a mix containing 30 mM EDTA and 0.7% SDS. Brief treatments by RNAse A and Proteinase K were followed by phenol-chloroform-isoamyl alcohol DNA extraction. The pellets were resuspended in 16 μl of TE and 4 μl of 5x loading buffer and only half of this solution was loaded on a 1% agarose gel in TAE 1x. The gels migrated at 55 V for 15 hr at 4°C and were then stained with ethidium bromide to visualize total DNA. Finally, gels were dried out and analyzed by Phosphorimager to visualize newly-synthetized DNA.

### yCAF1-PCNA pulldown

Purified full-length yCAF1, or yCAF1V complexes were mixed with equal amounts of pure trimeric PCNA (10 μM each) in pulldown buffer (50 mM Tris pH 7.5, 500 mM NaCl) and left for 10 min on ice before incubation with 20 μL equilibrated FLAG beads for 1 hr at 4°C (shaking). Unbound material was removed and the beads were washed three times with 100 μL pulldown buffer. Bound protein was eluted with 2 × 30 μL FLAG peptide (0.4 mg/ml) in 20 mM Tris pH 7.5, 300 mM NaCl and 10 min incubation at room temperature each time. Eluted proteins were analyzed by SDS-PAGE and western blot using standard procedures.

For western blot analysis, the samples were separated by SDS-PAGE and transferred to a nitrocellulose membrane, confirmed by Ponceau S red staining. The membrane was blocked with 5% defatted milk in TBST and incubated with monoclonal mouse anti-PCNA antibody (1:4000, Abcam, UK) in TBST over night at 4°C. After washing with TBST, the blot was incubated with HRP-conjugated anti-mouse secondary antibody (1:10 000, Sigma, St. Louis, MO) in TBST for 1 hr at room temperature. Finally, PCNA was detected by chemiluminescent signal from the ECL Prime Western Blotting Detection Reagent (Amersham). Subsequently, the membrane was stripped using 0.2 M glycine, pH 2.2, 0.1% SDS and 1% Tween using standard procedures. After incubation with anti-FLAG antibody (1:1000, Sigma) in TBST for 1 hr at room temperature the membrane was further treated as described for anti-PCNA.

## Acknowledgements

We thank Paul Kaufman for donating the plasmids pPK133, pPK160 and pPK134. Luca Signor (IBS, Grenoble) for mass spectrometry analysis. Adam Round and Martha Brennich (EMBL Grenoble) for support with SAXS analysis. Correspondence and requests for materials should be addressed to DP This work used the platforms of the Grenoble Instruct centre (ISBG; UMS 3518 CNRS-CEA-UJF-EMBL) with support from FRISBI (ANR-10-INSB-05–02) and GRAL (ANR-10-LABX-49–01) within the Grenoble Partnership for Structural Biology (PSB). This work was supported by the ANR Grant 'Replicaf' (ANR-16-CE11-0028-02) to FO, GA and DP. This work was supported by la Ligue Nationale contre le Cancer (Equipe labellisée Ligue), ANR-11-LABX-0044_DEEP and ANR-10-IDEX-0001–02 PSL, ANR-12-BSV5-0022-02 'CHAPINHIB', ANR-14-CE16-0009 'Epicure', ANR-14-CE10-0013 'CELLECTCHIP', EU project 678563 'EPOCH28', ERC-2015-ADG- 694694 'ChromADICT', ANR-16-CE15-0018 'CHRODYT', ANR-16-CE12-0024 'CHIFT'. DL was supported by the University Paris Sud and DS was supported by PSL.

## Additional information

### Funding

| Funder | Grant reference number | Author |
| --- | --- | --- |
| Agence Nationale de la Recherche | ANR-16-CE11-0028-02 | Paul Victor Sauer<br>Jennifer Timm<br>Danni Liu<br>David Sitbon |

The funders had no role in study design, data collection and interpretation, or the decision to submit the work for publication.

### Author contributions

PVS, Data curation, Investigation, Writing—original draft, Writing—review and editing; JT, Data curation, Investigation, Writing—review and editing; DL, Data curation, Investigation; DS, NM, JL, Data curation, Formal analysis, Writing—review and editing; EB-E, Data curation, Formal analysis, Investigation, Writing—review and editing; CV, Data curation, Formal analysis; FO, Data curation, Funding acquisition, Investigation, Project administration, Writing—review and editing; GA, Funding acquisition, Project administration, Writing—review and editing; DP, Conceptualization, Supervision, Funding acquisition, Writing—original draft, Project administration, Writing—review and editing

### Author ORCIDs

Paul Victor Sauer, http://orcid.org/0000-0001-7204-5863
Daniel Panne, http://orcid.org/0000-0001-9158-5507

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
