## [Decision Letter]

Thank you for submitting your article "Insights into the molecular architecture and histone H3-H4 deposition mechanism of the Chromatin assembly factor 1" for consideration by *eLife*. Your article has been reviewed by three peer reviewers, and the evaluation has been overseen by a Reviewing Editor and John Kuriyan as the Senior Editor. The reviewers have opted to remain anonymous.

The reviewers have discussed the reviews with one another and the Reviewing Editor has drafted this decision to help you prepare a revised submission.

Summary:

The reviewers find that the paper describes novel aspects of the mechanism of histone (H3-H4)_2_ tetrasome deposition by Chromatin Assembly Factor-1, particularly in regard to the role of DNA binding in the reaction.

Central conclusions:

1) There is a previously unrecognized DNA binding domain in the Cac1 subunit of yeast CAF-1 that is distinct from the winged-helix domain (WHD) near the C-terminus.

2) This DNA binding domain is required for efficient histone deposition in vitro.

3) Under the conditions used here, the CAF-1 complex interacts with a H3-H4 dimer, rather than a tetramer as suggested by previous studies.

Essential revisions:

1) There is insufficient comparison of these results to previous studies of the stoichiometry of histone binding, which came to a different conclusion. For example, in Figure 1, the size exclusion profile of CAF-1 and CAF1-H3-H4 complex is totally different from previous data. Winkler et al. 2012 show clear separation between CAF-1 and CAF1-histone complexes via S200 size exclusion chromatography. The tetramerization of H3-H4 is sensitive to ionic strength, and it must be noted that the buffer conditions used here, especially with regard to ionic strength, were considerably different in some cases to those that have been previously published. For example, Winkler et al. 2012 carry out SEC-MALS under 150 mM NaCl and find that yCAF1 binds a tetramer of H3-H4, whereas in this present study 500 mM NaCl was used for SEC-MALS and SAXS, 200 mM NaCl used for AUC and 250 mM ammonium acetate used for native-MS. Is there any significance to the choices of conditions used here? Did the authors test what oligomerization state the free H3-H4 was in under the given assay conditions in MALS, AUC and Mass Spec? Because of these discrepancies, the authors need to test how differences in experimental conditions might affect the results.

2) Figure 3 presents a DNA binding role for the N-terminal region of CAF1 (incorporating the K/E/R domain), distinct from the winged-helix domain that has previously been characterized (Zhang, Gao et al. 2016). This is potentially novel, however, there are a number of serious concerns regarding these experiments:

A) The CAF1-DNA complex appears to migrate very close to the wells, and therefore may represent an aggregate rather than a complex with a defined stoichiometry and molecular weight.

B) The EMSA buffer contained 500 mM NaCl, which is sufficient to break the interaction between most DNA binding domains and their DNA substrate – which may be why no DNA binding is seen for yCAF1T.

C) The increase in Hill coefficient for yCAF1U and yCAF1X (Table 4) is not explained in terms of the proposed model.

D) The errors bars from the quantitation are very large.

E) The gel from the 10 bp ladder titration experiment is cropped so that the reader can't see the CAF1-DNA complexes (presumably a CAF1 bound DNA ladder exists further up the gel).

F) In the same experiment the titration is stopped whilst 'binding' of the 20 and 30 bp fragments is still continuing.

G) Binding is quantified as the disappearance of the free DNA fragments rather than the appearance of a shifted CAF1-DNA complex.

Without defining the CAF1-DNA complex, an equally valid interpretation of the experiments could be that CAF1 forms non-specific aggregates with DNA, with smaller DNA fragments showing less tendency to aggregate than larger DNA fragments. The aggregates have a large molecular weight and migrate close to the loading wells. Truncation of Cac1 to its globular core (CAF1V and T) results in a loss of this aggregative tendency, whilst successive truncation of the N-terminal domain (yCAF1U and X) mitigates aggregation. Therefore, there needs to be more consideration of the possibility that some of the CAF-1-DNA associations might reflect non-specific aggregation, so SEC-MALS characterization of CAF-1+147bp DNA mixtures needs to be done. Additionally, it is important to show the full gel for Figure 3 (DNA ladder binding experiment), indicating where the wells are located, and that further titration points (past 11.5 μM of CAF1) are added to the experiment to demonstrate that DNA <30 bp is not being bound by CAF1.

3) The data regarding the activity of the mutant CAF-1 complexes in the *Xenopus* extracts does not distinguish the activities or mechanisms of these, they all appear similarly inactive.

---

## [Author Response]

*Essential revisions:*

*1) There is insufficient comparison of these results to previous studies of the stoichiometry of histone binding, which came to a different conclusion. For example, in Figure 1, the size exclusion profile of CAF-1 and CAF1-H3-H4 complex is totally different from previous data. Winkler et al. 2012 show clear separation between CAF-1 and CAF1-histone complexes via S200 size exclusion chromatography. The tetramerization of H3-H4 is sensitive to ionic strength, and it must be noted that the buffer conditions used here, especially with regard to ionic strength, were considerably different in some cases to those that have been previously published. For example, Winkler et al. 2012 carry out SEC-MALS under 150 mM NaCl and find that yCAF1 binds a tetramer of H3-H4, whereas in this present study 500 mM NaCl was used for SEC-MALS and SAXS, 200 mM NaCl used for AUC and 250 mM ammonium acetate used for native-MS. Is there any significance to the choices of conditions used here? Did the authors test what oligomerization state the free H3-H4 was in under the given assay conditions in MALS, AUC and Mass Spec? Because of these discrepancies, the authors need to test how differences in experimental conditions might affect the results.*

We have attempted to reproduced the results shown by Winkler et al. 2012 and found that these conditions (20 mM Tris pH 7.5, 300 mM KCl, 5% glycerol and 1 mM TCEP) repeatedly resulted in poor sample behavior as judged by the higher polydispersity of the eluting complex and multiple ‘shoulders’ when using a >90% pure sample (see Author response image 1). We repeatedly obtained (as also shown in Figure 1) monodisperse CAF1 and CAF1-H3-H4 complexes in 500 mM NaCl and consistently found that the histone-bound complexes almost elute at the same position as CAF1 alone. Further optimization of the buffer conditions for CAF1 was performed using standard thermofluor assays which can be used to assess sample stability.

Of note, Liu et al. 2012 also analyzed the CAF1 and CAF1-H3-H4 complex by gel filtration in buffer containing 300 mM NaCl and found, in agreement with our results, the two complexes eluting a similar position from the column (Liu et al., 2012). Thus, conditions used by Winkler et al. 2012 appear to adversely affect sample behavior. Any determination of mass and stoichiometry under such non-optimal conditions is of course difficult (see table in Figure 8).

Author response image 1.**DOI:**
http://dx.doi.org/10.7554/eLife.23474.023

To validate our results, we used methods that allow mass determination independently of shape and matrix interactions such as AUC and native-MS. For both techniques, we used buffer conditions that allowed us to obtain consistent results with the least amount of observed aggregation. Obtaining consistent results with different techniques and buffer conditions supports the validity of our results.

To further address the reviewers concern, we have done the following experiments:

1) Native MS experiments of free H3-H4 in 1M ammonium acetate – shown in Figure 1—figure supplement 2.

2) Native MS of H3-H4 when supplied in excess to CAF1 in 250mM ammonium acetate – Figure 1—figure supplement 2.

3) SEC-MALS of yCAF1 at 20 mM Tris pH 7.5, 300 mM KCl, 5% glycerol and 1 mM TCEP (conditions of Winkler et al. 2012) (peaks polydisperse). See Figure 8.

*2) Figure 3 presents a DNA binding role for the N-terminal region of CAF1 (incorporating the K/E/R domain), distinct from the winged-helix domain that has previously been characterized (Zhang, Gao et al. 2016). This is potentially novel, however, there are a number of serious concerns regarding these experiments:*

A) The CAF1-DNA complex appears to migrate very close to the wells, and therefore may represent an aggregate rather than a complex with a defined stoichiometry and molecular weight.

On 6% PAGE gels, the CAF1-DNA complex migrates closed to the well but it is not located in the well, as would be expected for an aggregate. To illustrate this better, we have analyzed the CAF1-DNA complex in a lower percentage (4%) PAGE gel (Figure 4—figure supplement 2). The CAF1-DNA complex migrates nicely into the gel showing that it is not an aggregate.

*B) The EMSA buffer contained 500 mM NaCl, which is sufficient to break the interaction between most DNA binding domains and their DNA substrate – which may be why no DNA binding is seen for yCAF1T.*

To address this comment, we have performed the EMSA with yCAF1T at 20 mM Tris pH 7.5, 100 mM NaCl and 1 mM DTT (buffer conditions used in Zhang et al. 2016 to determine the DNA binding affinity of the isolated winged-helix domain of Cac1). We observed no obvious DNA binding as shown in Figure 9.

Author response image 2.**DOI:**
http://dx.doi.org/10.7554/eLife.23474.024

C) The increase in Hill coefficient for yCAF1U and yCAF1X (Table 4) is not explained in terms of the proposed model.

yCAF1U and yCAF1X mutants show clearly reduced DNA binding affinity as compared to yCAF1. As a result, the binding isotherms are more variable and show higher error bars as compared to yCAF1. It is not possible to reliably quantify kinetic parameters from these data to clearly distinguish between Hill coefficients of 2-4. It is however clear that the binding isotherms do not fit to a Hill coefficient of 1 (non-cooperative binding). Even in some of the best studied systems such as hemoglobin, in which four oxygen molecules are known to bind with positive cooperativity, the measured Hill coefficient ranges from 1.7 to 3.2 rather than 4 (Weiss, 1997).

As there are a number of binding sites available on the extended DNA substrate, the Hill coefficient in this case provides more an ‘interaction’ coefficient reflecting cooperative binding rather than as a reliable estimate of binding sites (Weiss, 1997). Considering these limitations, we have not attempted to explain the increase in Hill coefficient in terms of a molecular model.

D) The errors bars from the quantitation are very large.

As indicated above, mutants that show reduced DNA binding affinity also show more variable binding isotherms. We do not think that this is unexpected and these errors reflect the precision of this EMSA experiment.

E) The gel from the 10 bp ladder titration experiment is cropped so that the reader can't see the CAF1-DNA complexes (presumably a CAF1 bound DNA ladder exists further up the gel).

Visualization of the entire range of the 10bp DNA ladder required a 10% native PAGE gel. CAF1-DNA complexes do no enter such high percentage gels and are seen in the loading well as shown in the uncropped gel in Figure 3—figure supplement 1.

F) In the same experiment the titration is stopped whilst 'binding' of the 20 and 30 bp fragments is still continuing.

We could not detect binding of short (15-30bp) DNA duplexes in our experiments. Only at very high CAF1 concentrations we observed some weak DNA binding. An example is shown below. Thus, while longer DNA substrates provide good CAF1 binding sites, short DNA substrates do not. We think this point is clearly illustrated in Figure 10.

Author response image 3.**DOI:**
http://dx.doi.org/10.7554/eLife.23474.025

*G) Binding is quantified as the disappearance of the free DNA fragments rather than the appearance of a shifted CAF1-DNA complex.*

It is more reliable to quantify DNA substrate depletion in these experiments. To clarify this point, we have included a statement in the legend of Figure 3: ‘CAF1- DNA binding was quantified by measuring DNA substrate depletion.’

*Without defining the CAF1-DNA complex, an equally valid interpretation of the experiments could be that CAF1 forms non-specific aggregates with DNA, with smaller DNA fragments showing less tendency to aggregate than larger DNA fragments. The aggregates have a large molecular weight and migrate close to the loading wells. Truncation of Cac1 to its globular core (CAF1V and T) results in a loss of this aggregative tendency, whilst successive truncation of the N-terminal domain (yCAF1U and X) mitigates aggregation. Therefore, there needs to be more consideration of the possibility that some of the CAF-1-DNA associations might reflect non-specific aggregation, so SEC-MALS characterization of CAF-1+147bp DNA mixtures needs to be done. Additionally, it is important to show the full gel for Figure 3 (DNA ladder binding experiment), indicating where the wells are located, and that further titration points (past 11.5 μM of CAF1) are added to the experiment to demonstrate that DNA <30 bp is not being bound by CAF1.*

The reviewers are correct in pointing out the possibility that CAF1 forms non-specific DNA aggregates as the yCAF1-DNA complex migrates close to the loading well under these conditions (6% Polyacrylamide gel). New experiments done on 4% Polyacrylamide gels show however that CAF1 forms a well-defined DNA complex that migrates into the gel (shown in Figure 4—figure supplement 2). We also show the full gel for Figure 3 as requested (Figure 3—figure supplement 1). As indicated above, in this experiment the CAF1-DNA complexes do not enter the gel due to the high percentage of polyacrylamide used.

We also have done CAF1-DNA binding experiments at higher CAF1 concentrations as requested. These data show that at very high concentrations of CAF1, shorter (17bp) DNA substrates can be bound by CAF1 (see Figure 3). However longer DNA substrates (42bp and 84bp) are bound with higher affinity (Figure 3). Analysis of the binding isotherms shows that long but not short DNA substrates show cooperative binding behavior (Figure 3).

We attempted to characterize the CAF1-DNA complex by SEC-MALS as requested, but found that the DNA and CAF1 did not co-elute from the SEC column. This is not surprising considering the weak DNA binding affinity of CAF1 (in the low μM range). Native PAGE analysis, as shown in Figure 3 is more suitable for the analysis of weakly interacting DNA binding complexes.

*3) The data regarding the activity of the mutant CAF-1 complexes in the Xenopus extracts does not distinguish the activities or mechanisms of these, they all appear similarly inactive.*

We agree with the reviewer’s comment and have addressed this point by carefully repeating the experiments to define the exact titration limit at which the wild type yCAF1 appears to be fully able to compensate for the endogenous xCAF1 loss. We used the obtained parameters to further analyze the activity of mutant CAF1 complexes in DNA synthesis-coupled nucleosome assembly. As shown in Figure 5, at higher ratios of mutant CAF1:DNA, some nucleosome assembly activity is recovered for CAF1T, U and X. We found that the CAF1T mutant has slightly higher activity as compared to CAF1U and X. Our interpretation is that while both the WHD and K/E/R-rich coiled-coil domains contribute to nucleosome assembly activity of CAF1, the WHD domain is slightly more important. CAF1V, the minimal CAF1 construct lacking both the WHD and the K/E/R-rich coiled-coil domain, is defective in nucleosome assembly activity. These new data are shown in Figure 4 and described in the subsection “DNA-binding of yCAF1 is required for DNA synthesis-coupled nucleosome assembly”.